# Antiviral fibrils of self-assembled peptides with tunable compositions

Joseph Dodd-o[1,15], Abhishek Roy[1,15], Zain Siddiqui[1], Roya Jafari[2], Francesco Coppola [2], Santhamani Ramasamy[3], Afsal Kolloli[3], Dilip Kumar[4], Soni Kaundal[4], Boyang Zhao[4], Ranjeet Kumar[3], Alicia S. Robang[5], Jeffrey Li[5], Abdul-Rahman Azizogli[6], Varun Pai [6], Amanda Acevedo-Jake[1], Corey Heffernan[1,7], Alexandra Lucas[8], Andrew C. McShan [9], Anant K. Paravastu [5], B. V. Venkataram Prasad [4], Selvakumar Subbian [3], Petr Král [2,10,11,12,16] ✉ & Vivek Kumar [1,6,7,13,14,16] ✉

The lasting threat of viral pandemics necessitates the development of tailorable first-response antivirals with specific but adaptive architectures for treatment of novel viral infections. Here, such an antiviral platform has been developed based on a mixture of hetero-peptides self-assembled into functionalized β-sheets capable of specific multivalent binding to viral protein complexes. One domain of each hetero-peptide is designed to specifically bind to certain viral proteins, while another domain self-assembles into fibrils with epitope binding characteristics determined by the types of peptides and their molar fractions. The self-assembled fibrils maintain enhanced binding to viral protein complexes and retain high resilience to viral mutations. This method is experimentally and computationally tested using short peptides that specifically bind to Spike proteins of SARS-CoV-2. This platform is efficacious, inexpensive, and stable with excellent tolerability.

In recent decades, many novel viruses originating in the animal kingdom have been spreading in the rapidly growing human population. Among them, the highly contagious severe acute respiratory syndrome coronavirus 2 (SARS-CoV-2), which caused the COVID-19 pandemic, has claimed millions of lives worldwide and overwhelmed global healthcare systems for several years[1]. During the infection process, the viral 'Spike' protein binds the angiotensin-converting enzyme 2 (ACE2) receptor that is expressed ubiquitously in human cells[2–4]. The mRNA vaccines developed against SARS-CoV-2 have proven to be an effective strategy against severe disease and death in infected patients, and whilst more effective against early viral strains; their protection against new viral variants becomes less efficient, unless they are based on new viral strains[5].

[1]Department of Biomedical Engineering, New Jersey Institute of Technology, Newark, NJ 07102, USA. [2]Department of Chemistry, University of Illinois at Chicago, Chicago, IL 60607, USA. [3]Public Health Research Institute, New Jersey Medical School, Rutgers University, Newark, NJ 07103, USA. [4]Department of Molecular Virology & Microbiology, Baylor College of Medicine, Houston, TX 77030, USA. [5]School of Chemical and Biomolecular Engineering, Georgia Institute of Technology, Atlanta, GA 30332, USA. [6]Department of Biological Sciences, New Jersey Institute of Technology, Newark, NJ 07102, USA. [7]SAPHTx Inc, Newark, NJ 07104, USA. [8]Center for Personalized Diagnostics and Center for Immunotherapy Vaccines and Virotherapy, Biodesign Institute, Arizona State University, 727 E Tempe, AZ, USA. [9]School of Chemistry and Biochemistry, Georgia Institute of Technology, Atlanta, GA 30332, USA. [10]Department of Physics, University of Illinois at Chicago, Chicago, IL 60607, USA. [11]Department of Pharmaceutical Sciences, University of Illinois at Chicago, Chicago, IL 60607, USA. [12]Department of Chemical Engineering, University of Illinois at Chicago, Chicago, IL 60607, USA. [13]Department of Chemical and Materials Engineering, New Jersey Institute of Technology, Newark, NJ 07102, USA. [14]Department of Endodontics, Rutgers School of Dental Medicine, Newark, NJ 07103, USA. [15]These authors contributed equally: Joseph Dodd-o, Abhishek Roy. [16]These authors jointly supervised this work: Petr Král, Vivek Kumar. ✉e-mail: pkral@uic.edu; vak@njit.edu

Recombinant hACE2 and virus-specific antibodies from convalescent patient plasma have been explored as decoys for SARS-CoV-2 inhibition, but the risk of blood-borne disease, immune rejection, and demand for logistical expertise to purify and manufacture biological (blood) products, limits broad availability and extends higher costs[6,7]. In general, the specificity of antibodies towards target proteins is conferred by the conformations that the complex antibody active sites adopt and their surface interactions. Despite their large sizes, numerous antibodies (convalescent plasma to in situ vaccine-derived antibodies) have gradually diminishing activities against viral variants/mutations[8,9].

The early presence of a high-resolution atomistic structure of SARS-CoV-2 bound to ACE2 has facilitated the rational design of novel therapeutics[10] targeting the Spike receptor-binding domain (RBD) interaction with host ACE2[2,3,6,11–16]. Based on the ACE2-RBD coupling, design strategies that employ the human ACE2 structure (binding motifs) have been identified and proposed as proteins and peptides for therapeutic development[17–22]. As promising peptide candidates display a lack of stability and live virus efficacy, molecular modeling studies were used to find site-specific amino acid mutations to optimize the helical strength, maintain low antigenicity, and have a high affinity for RBD[17,18,23]. A strategy to target smaller antigenic determinants with shorter proteins/peptides may confer variant specificity, at the cost of specificity restricted to a small binding region. However, self-assembled peptide (SAP) conjugates may enhance binding[24,25] to a target through non-covalent stabilization by multivalent[25–36] supramolecular interactions[37]. At the same time, larger self-assembled constructs can form a supramolecular assembly atop multiple RBDs at the viral surface, thereby inhibiting viral proteins from binding to cell receptors. This strategy is premised on studies that have investigated SAP-like peptides with multivalent tailorable antigen presentation[38] in self-adjuvanting vaccines[39,40] for upregulated targets in cancer[41,42], and other pathogens[40,43].

Here, we introduce a tunable and scalable antiviral platform that utilizes the specificity of peptides in their binding to receptors, which is strengthened by multivalency introduced through a charged amphiphilic domain that mediates the self-assembly of the peptides. Here, we have designed and tested functionalized hetero-peptides that can self-assemble into fibers and bind to SARS-CoV-2 Spike-RBDs, and perform facile tuning of the peptide sequence and secondary structure composition through rationalized computational design demonstrated for pan-coronavirus targeting. The conjugation of the functional Spike-binding peptides (SBP1/2/3) to an SAP domain preserves the in silico RBD affinity of the constructs (Fig. 1A–F). Moreover, SAP conjugated peptide inhibitors (ESBP1/2/3) were hypothesized to bind viral Spike and self-assemble atop the viral particles, effectively inhibiting the Spike from binding with ACE2 (Fig. 1G). Interestingly, the SAP domain alone has non-specific (ionic) interactions with Spike (demonstrated in silico and in vitro), and additionally, significantly, synergize RBD targeting SAP peptides in mixtures against live virus.

## Results and discussion

The published atomistic binding structure of Spike-RBD coupled to ACE2[44] provided a means to understand the interaction between the ACE2 α1 helix domain and Spike-RBD, which was mutated to enhance its binding to RBD (Fig. 1). These mutated peptides demonstrated that a more stable α-helix that aided in binding to RBD, with nM inhibition of SARS-CoV-2[23], akin to other ACE2 domain mimicry strategies recently investigated[45–47].

### Enhanced binding with self-assembling sequences

Initial attempts in peptide design involved isolation of a short Spike-binding peptide (SBP) sequence (IEEQAKTFLDKFNHEAEDLFYQS) from ACE2 α1[17]. This short sequence, termed SBP1, was the first bioactive domain that was tested. The SAP domains were conjugated to SBP1 with 2 alternatively charged SAP terminal residues, glutamic acid E[24,48] and lysine K[30,31,49] (Fig. 2A). We observed a significant (~70%) inhibition of pseudovirus by SBP1 from media control (Fig. 2B), as reported previously[17,18]. E-flanked SAP termed E1 (E-SLSLSLSLSLSL-E) was conjugated to SBP1 (ESBP1), which inhibited pseudovirus to a similar degree, while K-flanked SAP termed K1 (K-SLSLSLSLSLSL-K)

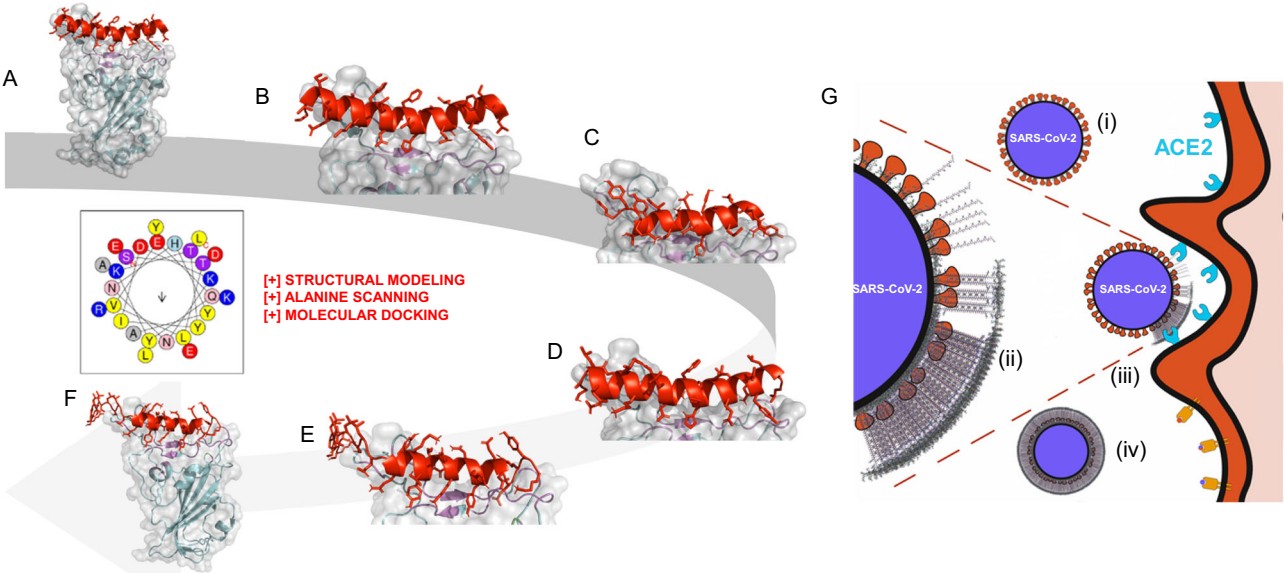

**Fig. 1 | Design strategy of self-assembling peptides (SAPs) as targeted inhibitors of SARS-CoV-2. A** ACE2 α1 helix (red) complexed on SARS-CoV-2 Spike RBD (gray surface, cyan & purple backbone) generated from PDB ID: 6M0J[69]. **B** Key residues highlighted in yellow were deemed most significant for the ACE2 α1 helix peptide as determined by BUDE Alanine Scanning, ΔΔG 52, while purple residues on the receptor highlight the binding pocket on the Spike RBD[52]. **C** Derived from ACE2 α1[18], SBP2 (mutated SBP1) docked on Spike-RBD[51]. **D** Further mutation informed by structural modeling, interacting residue identification[52], and molecular docking yielded SBP3. **E** Close up of the interaction of ESBP3 monomer with Spike-RBD. **F** Full view of the ESBP3 monomer−Spike-RBD complex. **G** We hypothesize that SBP-functionalized SAP monomers comprise a target domain that: (i) binds Spike of SARS-CoV-2, (ii) self-assembles atop the viral particle, (iii) inhibits virus binding to ACE2, and (iv) coats the virus for clearance by the body.

**A**

| Name | SAP domain | Linker | Targeting | Ref. |
|---|---|---|---|---|
| SBP1 | | | IEEQAKT**FLD**KFNHEAE**D**LF**YQ**S | 17 |
| ESBP1 | **E**SLSLSLSLSLSL**E** | G | IEEQAKT**FLD**KFNHEAE**D**LF**YQ**S | |
| KSBP1 | **K**SLSLSLSLSLSL**K** | G | IEEQAKT**FLD**KFNHEAE**D**LF**YQ**S | |

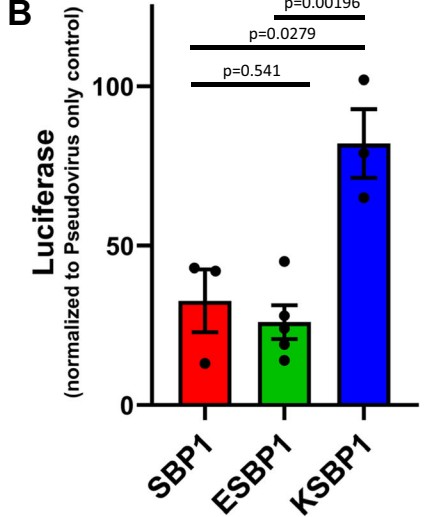

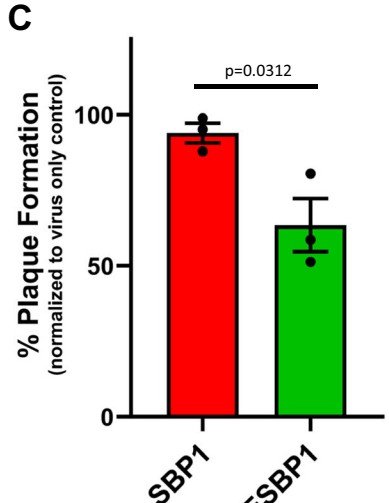

**Fig. 2 | Evaluation of Spike-binding peptides and anti-SARS-CoV-2 SAPs.**
**A** Spike-binding proteins (SBP) derived from truncated ACE2 peptidase domain's α1 helix[17]. SBP1 appended to the termini of negative and positive SAP domains (E-SLSLSLSLSLSL-E or K-SLSLSLSLSLSL-K), yielding **E**SBP1 and **K**SBP1. The bold-underlined residues are essential for Spike binding and conserved in all constructs. **B** Constructs were screened using a Spike expressing inactivated pseudovirus luciferase reporting assay to compare the ability of SBP1, ESBP1, and KSBP1 (10 μM) to inhibit viral infection of 293 T cells overexpressing ACE2. $n = 3$ samples (KSBP1),

$n = 3$ (control, SBP1) or 5 samples (ESBP1). Data are presented as mean ± SEM. One-way ANOVA ($df = 2$, F-statistic = 13.6, $p = 0.00267$; $p$ values from pairwise $t$ test between groups shown). **C** Antiviral efficacy of SBP1 and ESBP1 (10 μM) using live SARS-CoV-2 virus. Data are presented as mean ± SEM; $p$ values from pairwise $t$ test between groups shown. Therapeutic inhibition of Infection in non-human primate (Vero) cells in presence of SBP1 and ESBP1 over control. Source data are provided in the Source Data file.

conjugated to SBP1 (KSBP1) had a significant decrease in inhibitory effect (Fig. 2B). A secondary B.1 lineage live virus plaque formation assay was performed with SBP1 and ESBP1 to confirm if the self-assembling domain could assist in viral inhibition. In the live virus assay, counts of infectious plaques confirmed a significant improvement in viral inhibition by ESBP1 relative to SBP1 alone (Fig. 2C).

During the testing of ESBP1 (2021), a novel Spike-binding domain that had a stabilized ACE2-based helix was discovered by Karoyan et al.[23]. The sequence was modified to include only canonical amino acids, yielding the bioactive sequence SBP2 (SALE-EQLKTFLDKFMHELEDLLYQLAL), which had a strong α-helical structure (Supplementary Fig. 1). Conjugation to the E1 self-assembling domain yielded a sequence ESBP2 (Supplementary Fig. 1). Notably, SBP2 and ESBP2 showed excellent cytocompatibility and inhibition of live virus. ESBP2 safety was evaluated in vivo via 10 days repeat multi-dose IV administration of ~1w% ESBP2 which showed no adverse effect in 225–250 g adult rats and ~4–8 hours clearance in acute 24-hour single-dose pharmacokinetic (PK) evaluation (Supplementary Fig. 1). However, the ESBP2 peptide did not form stable β-sheets, as evidenced by an α-helical signature in CD (Supplementary Fig. 1C) and visualized aggregates by AFM (Supplementary Fig. 1D). Notwithstanding, this supports our hypothesis of SAP functionalization of binding domains, specifically demonstrating that SAPs with E termini can enhance Spike binding and inhibition (Fig. 2A, B).

**Conformational optimizations of peptides improve fibrillation**
Docking of SBP2 to Spike showed a HADDOCK[50,51] score inferior to unmodified ACE2 α1 helix. Further optimization of SBP2[23] revealed the importance of Tyr or homo-Tyr mutations for Leu-7 in their 20–30-mer peptides. Bude Alanine Scanning (BalaS)[52] of the resulting lowest

energy pose identified end residues for truncation and central residues for modification. This sequence, SBP3 (QYKTYIDKNNHYAEDERYK, Fig. 3A, B), showed improved Spike binding in silico (Fig. 3C). We noted that the smaller size of this sequence may allow less steric hindrance in self-assembled structures. Circular dichroism of E1 conjugated SBP3 (ESBP3, Fig. 3A, B) showed predominant β-sheet conformation. ESBP3 showed the formation of fibrils in AFM (Supplementary Fig. 2) and negatively stained TEM (Fig. 3E), which appear to form aggregates radially on the Spike RBD (Fig. 3F). ESBP3 further demonstrated dose-dependent inhibition of live virus.

Thermal signatures from isothermal titration calorimetry (ITC) supported spontaneous self-assembly of ESBP3 monomers and ESBP3-Spike-binding events (Supplementary Fig. 2). ESBP3 showed excellent stability in lyophilized form as well as in formulation at a variety of temperatures over 12 weeks (Supplementary Fig. 2). Like ESBP2, ESBP3 showed excellent cytocompatibility with human alveolar epithelial A549 cells and dose-dependent antiviral inhibition with a calculated $IC_{50}$ of 2.5 μM (Supplementary Fig. 1, Fig. 3G). ESBP3 showed rapid clearance from the circulation (<4 hours) after IV bolus administration—characteristic of IV (peptide) therapeutics (Supplementary Fig. 2). Daily repeated IV dosing for 10 days showed no adverse effect in rodents. Subcutaneously implanted boluses in rats and repeated daily intranasal (IN) dosing in mice showed no significant body weight changes or adverse organ morphology (Supplementary Fig. 2). These 2 routes represented safety for potential instillation routes for prophylaxis in the nasal passageways and potential treatment IV. These biocompatibility results support safety but may warrant future studies that probe interactions of peptides and their maintained efficacy when in contact with mucosal/plasma fluids.

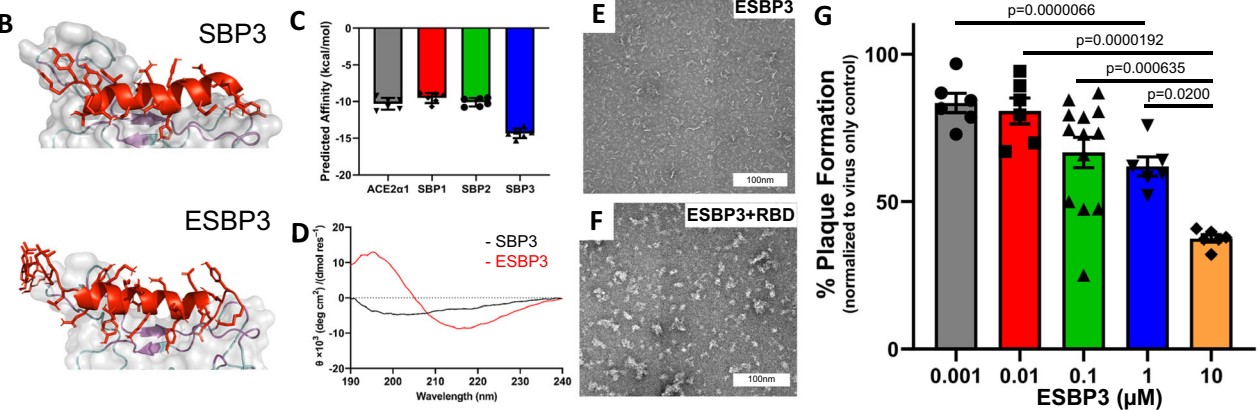

| Name | SAP domain | Linker | Targeting |
|---|---|---|---|
| SBP1 | | | IEEQAKT**FLD**KFNHEAE**D**LF**YQ**S |
| SBP2 | | | SALEEQYKT**FLD**KFMHELE**D**LL**YQL**AL |
| SBP3 | | | QYKT**YID**KNNHYAE**D**ER**YK** |
| ESBP3 | ESLSLSLSLSLSLE | G | QYKT**YID**KNNHYAE**D**ER**YK** |
| E1 | ESLSLSLSLSLSLE | | |

**Fig. 3 | Development of Spike-binding sequence. A** Sequence comparison of SBP1 and SBP2 to yield a truncated and optimized SBP3 and ESBP3. The bold-underlined residues are essential for Spike binding and generally conserved in all constructs **B** Comparative analysis of SBP3 and ESBP3 (both shown in orange) binding of Spike protein (shown in green) by in silico modeling. **C** PRODIGY score analysis of SBP3 for Spike RBD relative to previously published sequences SBP1 and SBP2. Data shown as Mean ± SEM. **D** CD analysis of secondary structure of SBP3 and ESBP3. **E** TEM highlighting short fibrillar structure formation in ESBP3 **F** TEM showing ESBP3 assembly around SARS-CoV-2 Spike RBD (Scale bar: 100 nm, ×40,000 magnification). **G** The ability of live SARS-CoV-2 virus to infect non-human primate (Vero) cells in the presence of ESBP3 in a dose-dependent fashion (mean infectious plaque counts, normalized to 'no peptide' control; source data are provided as a Source Data file). $n = 6$ samples (0.001, 0.01, 1, and 10 μM) or 13 samples (0.1 μM condition). Data are presented as mean ± SEM. One-way ANOVA ($df = 2$, F-statistic = 12.1, $p = 0.00000421$; Tukey Honestly Significant Difference (HSD) post-hoc test for multiple comparisons). Source data are provided in the Source Data file.

## Examination of the self-assembled β-sheets

To better understand the structure of these mimics, various self-assembled peptide structures were analyzed with molecular dynamics (MD) simulations. First, we examined whether parallel or antiparallel orientations take part between E1 self-assembled in β-sheet planes. The calculated MM-GBSA free energies (see Methods for details) of binding between E1 and its surrounding β-sheet plane showed that E1 preferentially assembles into an antiparallel β-sheet. The calculated free energies were ΔG = −88.45 kcal/mol, −73.37 kcal/mol, and −108.10 kcal/mol for all parallel, planar antiparallel, and all antiparallel, respectively (Fig. 4A–C). This suggests a preponderance towards fully antiparallel self-assembly, but not exclusively antiparallel arrangement; statistical distributions of peptide configurations within the self-assembled fibrils are controlled by their free energies. The MD simulations revealed variable non-periodic twists within the fibrils, analogous to variable twist rates observed within an E1 fiber during Cryo-EM image collection. This variability in twist limited our ability to resolve the cryo-EM model at adequate resolution, but it did allow us to observe spacing between the monomers and an approximate multimeric structure similar to our models (Supplementary Fig. 3).

Motivated by the improved structure of ESBP3 by site-specific mutations, we investigated, by MD simulations, the predicted reduction in steric hindrance by incorporating just the self-assembling domain E1 into ESBP3 fibers. We generated models of 16-mer fibers containing a 7:1 molar ratio and a 3:1 molar ratio of E1 and ESBP3, and pure ESBP3 (Fig. 4D–F). The E1 fibers containing no bioactive domain were hypothesized to serve as a spacer molecule for the multidomain ESBP3 fibers. Accessible Surface Area (ASA) measurements showed an inverse relationship between the ESBP3 molar proportion and the area of mimic accessible. This agrees with the calculated average ASA of 2761 Å²/mimic for 7:1 E1:ESBP3, 2614 Å²/mimic for 3:1 E1:ESBP3, and 2319 Å²/mimic for all ESBP3 (Fig. 4D–F). This warranted further testing through solid-state NMR interactions and, ultimately, live virus inhibition.

When the E1, ESBP3, and E1:ESBP3 in a 3:1 molar ratio assembly were examined with NMR, we observed centrifuge pellets and concomitant solid-state NMR signals from the samples of E1 and 3:1 E1:ESBP3. This indicates their assembly form organized secondary structures typical for a peptide nanofiber assembly, vs ESBP3 (no pellet/no solid-state NMR signal). Comparing the spectra collected for the E1 and 3:1 E1:ESBP3 assemblies, both exhibited peaks with linewidths of ~2 ppm, some overlapping peaks, and some peaks that are unique to the 3:1 E1:ESBP3 sample (Fig. 4G). The linewidths were typical for $^{13}C$ natural abundance spectra collected for amyloid fibrils of designer peptides[53]. While natural abundance $^{13}C$ NMR is typically insufficient to perform spectral assignments (correspondence between NMR peaks and $^{13}C$ sites), partial $^{13}C$ peak assignments were possible because known $^{13}C$ chemical shift ranges for E, S, and L residues within the E-$(SL)_6$-E self-assembling domain in both peptides[54]. Many peaks observed in the E1 spectrum align with the peaks from 3:1 E1:ESBP3, as could be anticipated since both peptides assemble into β-sheets, and the E1 peptide is the most abundant peptide in both samples (Fig. 4G). The strongest signal was at 28 ppm, indicating that the presence of the ESBP3 peptide affects the assembled nanofiber structure, consistent with the SBP3 peptide domain associated with 25% of molecules in the 3:1 E1:ESBP3 sample (Fig. 4G). The signals indicate that the SBP3 domain was incorporated into the assembly, but also suggest that a portion of the residues in the SBP3 domain may be integrated into the β-sheet assembly.

Given the lacking solid-state NMR signal indicating the assembly of ESBP3 peptide relative to the E1:ESBP3 mixture, we sought to probe soluble monomeric and oligomeric peptide assemblies using solution

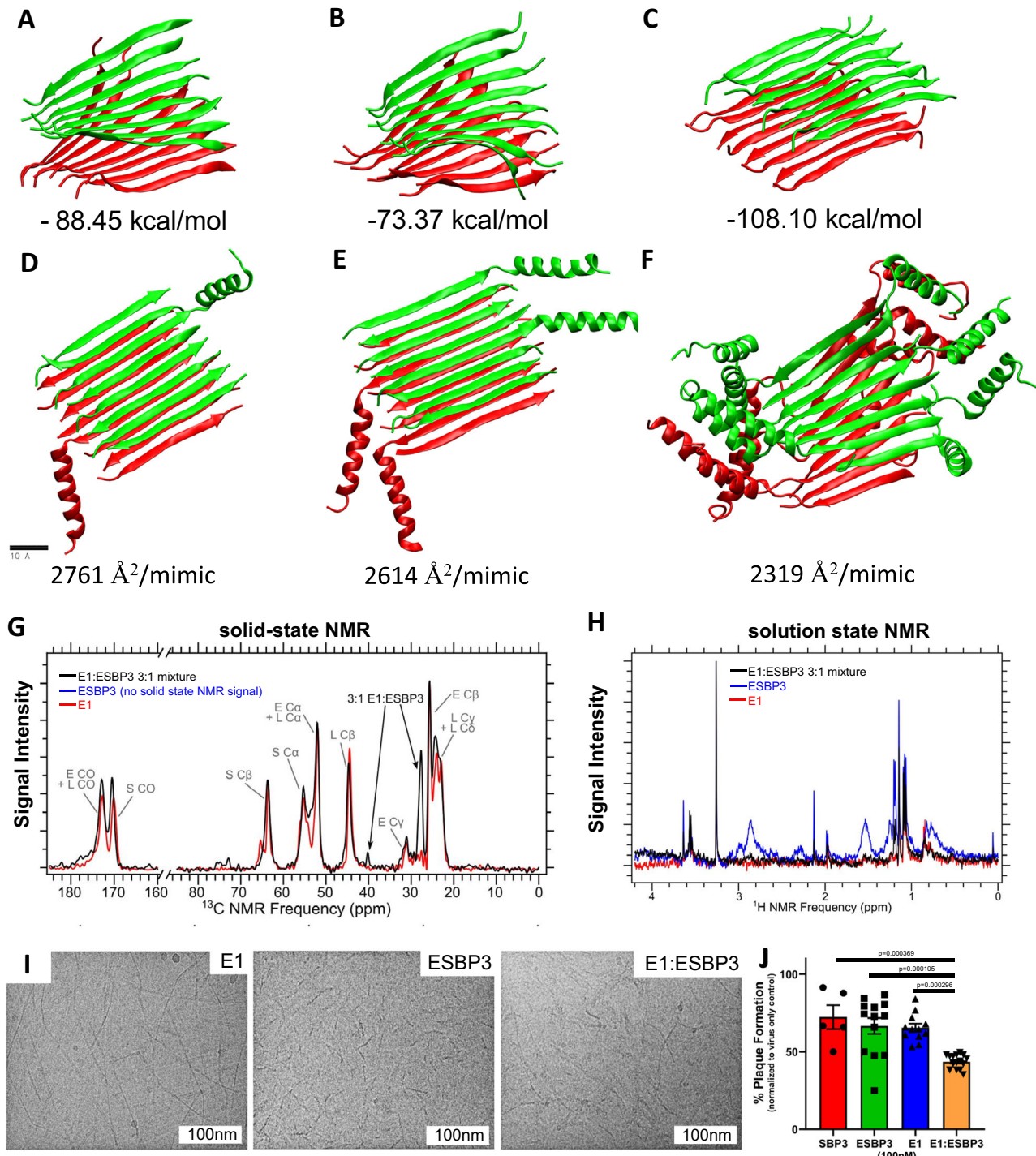

**Fig. 4 | Self-assembly of E1 and ESBP3 SAP.** MD simulations were performed to predict E1 fibril preponderance towards **A** all parallel, **B** planar antiparallel, and **C** all antiparallel β-sheets. MM-GBSA free energy of coupling between parallel fibers and antiparallel fibers indicates a more favorable assembly in antiparallel fashion. Combinations of **D** 1:7 ESBP3:E1, **E** 1:3 ESBP3:E1, **F** and all ESBP3 were simulated in β-sheet assemblies for 400 ns to assess the relationship between multidomain peptide concentration and interference between the bioactive domains. Accessible surface area (ASA) calculations of the MD trajectories indicate an inversely proportional relationship between multidomain peptide concentration and ASA. **G** Solid-state NMR spectra (1H-13C CPMAS, natural isotopic abundance peptide at 10 mg/mL) of ultracentrifuge pellets of E1 and 3:1 E1:ESBP3 assemblies. The ESBP3 solution produced no ultracentrifuge pellet such that solid-state NMR spectra could be collected. **H** 1D 1H solution NMR spectra of 1 mg/mL nature isotopic abundance peptide solutions of E1, ESBP3, and 3:1 E1:ESBP3 mixtures. **I** The fibrillar structures are clearly visible in vitreous-ice cryo-EM. **J** Virus-induced plaque inhibition on Vero cells shows differential efficacy in virus sequestration by E1:ESBP3 in a 3:1 ratio relative to E1 or ESBP3 alone. $n = 5$ samples (SBP3), $n = 12$ (E1), $n = 13$ (ESBP3) or $n = 15$ samples (E1:ESBP1). Data are presented as mean ± SEM. One-way ANOVA ($df = 3$, F-statistic = 25.5, $p = 0.0000104$; Tukey Honestly Significant Difference (HSD) post-hoc test for multiple comparisons). Source data are provided in the Source Data file.

NMR. To this end, 1D ${}^1$H solution NMR spectra of 1 mg/mL solutions of E1, ESBP3, and 3:1 E1:ESBP3 mixtures were acquired and compared (Fig. 4H). ESBP3 peptide exhibited a strong solution NMR signal (Fig. 4H), in stark contrast to its undetectable solid-state NMR signal (Fig. 4G). E1 peptide and 3:1 E1:ESBP3 mixtures exhibit very weak solution NMR signal (Fig. 4H) and corroborating solid-state NMR signals suggesting these peptides form nanofiber structures (Fig. 4G). Notably, the E1 peptide and 3:1 E1:ESBP3 mixtures do exhibit some signal in solution NMR experiments, albeit a different spectral profile from isolated ESBP3 peptide. Several NMR peaks in solution NMR spectra of ESBP3 exhibit narrow linewidths (<0.1 ppm full-width half maximum) consistent with unassembled, free peptide monomers in aqueous solution. Broad NMR peaks with wider linewidths (~0.2 ppm) were also present in the ESBP3 solution NMR spectra, indicating the presence of soluble, oligomeric, peptide assemblies (Fig. 4H)−these fibrillar structures were further visualized in cryo-EM (Fig. 4I). To determine whether the broad NMR peaks correspond to soluble peptide assemblies, we performed 1D ${}^1$H Carr-Purcell-Meiboom-Gill (CPMG) solution NMR experiments[55], which utilize optimized pulses to filter out NMR signals corresponding to larger molecules with unique relaxation properties (specifically, small $T_2$ relaxation values for oligomers with large molecular weights)[56]. The ${}^1$H CPMG experiments indeed illustrated the presence of both soluble, oligomeric assemblies and monomeric peptides of ESBP3, as shown by the loss of NMR signal corresponding to broad NMR peaks and the relatively unaffected NMR signal of the narrow NMR peaks as a function of the CPMG relaxation filter (Supplementary Fig. 4).

Taken together, solid- and solution-state NMR indicate that the SBP3 domain can be incorporated into a self-assembled nanostructure formed by the E1 domain. For the ESBP3 peptide solution, which contains E1 and SBP3 in every peptide molecule, the SBP3 interferes with E1 assembly, resulting in soluble aggregates that can be detected by solution NMR, but did not pellet via ultracentrifugation. The solid-state NMR data showed that ESBP3 was incorporated into a nanostructure that could be pelleted when ESBP3 was co-assembled with E1 in the 3:1 E1:ESBP3 sample (Fig. 4G, H). A comparison of the spectrum from 3:1 E1:ESBP3 to the spectrum of an E1 assembly indicates that SBP3 affects the assembled structure, but further research is necessary to pin down the structural details. Thus, despite a degree of efficacy we suggest that there may be room for improvement in the self-assembling peptide design. ESBP3 is limited in its ability to form β-sheets and fibrils alone (Figs. 3E, 4I), but it is stabilized into a fiber by doping with E1.

## Live virus inhibition with self-assembled peptide mimics

Live virus plaque inhibition assays against the B.1 strain were performed on E1, SBP3, ESBP3, and E1:ESBP3 in a 3:1 molar combination to determine their therapeutic efficacy. While no major differences were determined between E1, SBP3, and ESBP3, the 3:1 molar combination of E1 and ESBP3 had significantly higher viral inhibition (Fig. 4J). This enhanced viral inhibition in diluted fibrils could have several possible origins, such as specificity to live virus/ Spike by targeting RBD (of ESBP3/ E1:ESBP3) or non-specific ionic interactions of poly-anionic E1. However, spacing of ESBP3 domains with just E1 SAP domains, as suggested by NMR, promoted better presentation of the binding domain to RBD.

To understand these experimental inhibition observations, we performed MD simulations of individual peptides and their assemblies coupled to RBDs in the Alpha (B.1.1.7) and Omicron (B.1.1.529) variants. Starting with the individual peptides binding to the Alpha receptor, the E1 domain moved away from the canonical binding pocket (Fig. 5A). SBP3 migrated around the binding pocket, while rotating and translating away from the traditional residues associated with binding to ACE2 (Fig. 5B), but ESBP3 showed more efficient with a limited migration from the canonical binding pocket (Fig. 5C).

ESBP3 also demonstrated higher coupling energy with an average of −69.9 kcal/mol, vs. −38.9 kcal/mol and −38.4 kcal/mol for SBP3 and E1 respectively. For the Omicron RBD, E1 localized around the positively charged RBD (Fig. 5D, Supplementary Figure 5) with a similar coupling energy of −32.0 kcal/mol. SBP3 showed a decreased coupling energy against Omicron vs. Alpha (Fig. 5E) at −36.3 kcal/mol, but ESBP3 again showed improved binding relative to E1 and SBP3 at −95.5 kcal/mol (Fig. 5F). MD simulations against a widespread variant (B.1.529) suggested ESBP3 and E1 would have improved binding against the receptor-binding motif (RBM), likely due to the increased positive electrostatic surface potential localized on the Omicron RBD (Fig. 5E−G, Supplementary Movies. 1-3)[57]. This data supports the hypothesis that combining the SBP3 Spike-Binding Peptide with the E1 SAP can enhance its binding to the receptor (Fig. 2, Fig. 5A−F, Supplementary Fig. 5).

We simulated fibrils with the 7:1 E1:ESBP3 molar ratio 16-mer, the 3:1 E1:ESBP3 molar ratio, and all ESBP3 combinations coupled to the Alpha and Omicron variants. For Alpha variant, we saw average coupling energies increased with increasing relative ESBP3 concentration, with values of −41.5 kcal/mol, −50.4 kcal/mol, and −61.4 kcal/mol for 7:1 E1:ESBP3, 3:1 E1:ESBP3, and all ESBP3 respectively. For the Omicron variant, there was a less pronounced difference in energy, with values of −88.8 kcal/mol, −54.2 kcal/mol, and −70.9 kcal/mol for 7:1 E1:ESBP3, 3:1 E1:ESBP3, and all ESBP3, respectively (Fig. 5H, I, K, L, Supplementary Figure 6, Supplementary Movies 1−3). While the 7:1 E1:ESBP3 showed selective binding to the canonical binding pocket (Fig. 5G, J, Supplementary Movie 3), as the molar ratio of ESBP3 increased, the selectivity of binding started to decrease at full occupancy of the bioactive domains (Fig. 5H, K, Supplementary Movies 1−3).

Coupling energy of the whole fibril alone did not explain our live virus observations that spacing the multidomain peptides would yield a more effective antiviral (Fig. 5G−L, Supplementary Figure 6). At full occupancy of the fibril by the bioactive domains, various other coupling regions bound to RBDs (Fig. 5I, L, Supplementary Movies 3). Overall, our MD simulations of fibrils with different ratios of E1:ESBP3 showed that as the proportion of ESBP3 increases, ASA decreases, leading to disordered presentation of the bioactive domains[58] (Figs. 4E, F and 5G−L). This, along with our live virus data (Fig. 4J), suggests a hypothesized optimization[59−61] for future hybrid constructs by appropriate combinations of pure self-assembling peptide and multidomain peptide monomers is warranted as evidenced by the significantly enhanced binding and live virus inhibition by E1:ESBP3.

We introduced tunable and scalable antiviral therapeutics based on suitable peptide domains conjugated to short peptides capable of self-assembling into functionalized β-fibrils. When such hetero-peptides, based on mutated ACE-α1, are self-assembled at different relative concentrations, they allow multivalent binding to Spikes of multiple SARS-CoV-2 variants. EM imaging suggests the aggregation of fibers atop viral particles, depending on the E1:ESBP3 relative molarities. Interestingly, E1 alone showed broad antiviral potential principally due to its net charge interactions with viral protein coats, as reported for several similar poly-anionic scaffolds for SARS-CoV-2 treatment[62−64]. Simulations of these dynamical therapeutics revealed that inter-peptide coupling within the self-assembled antivirals might reduce the strength of multivalent binding to the viral receptors. This work provides a versatile platform with the potential to generate highly targeted and inexpensive drugs readily amenable to conventional drug fill-and-finish processes, which can be useful for large-scale production of this class of medicines.

## Methods

### Evaluation of SBP1 and SBP2

SBP1 sequence (IEEQAKTFLDKFNHEAEDLFYQS) was derived from the interacting regions of ACE2 by Pomplun et al.[17] SBP2 sequence

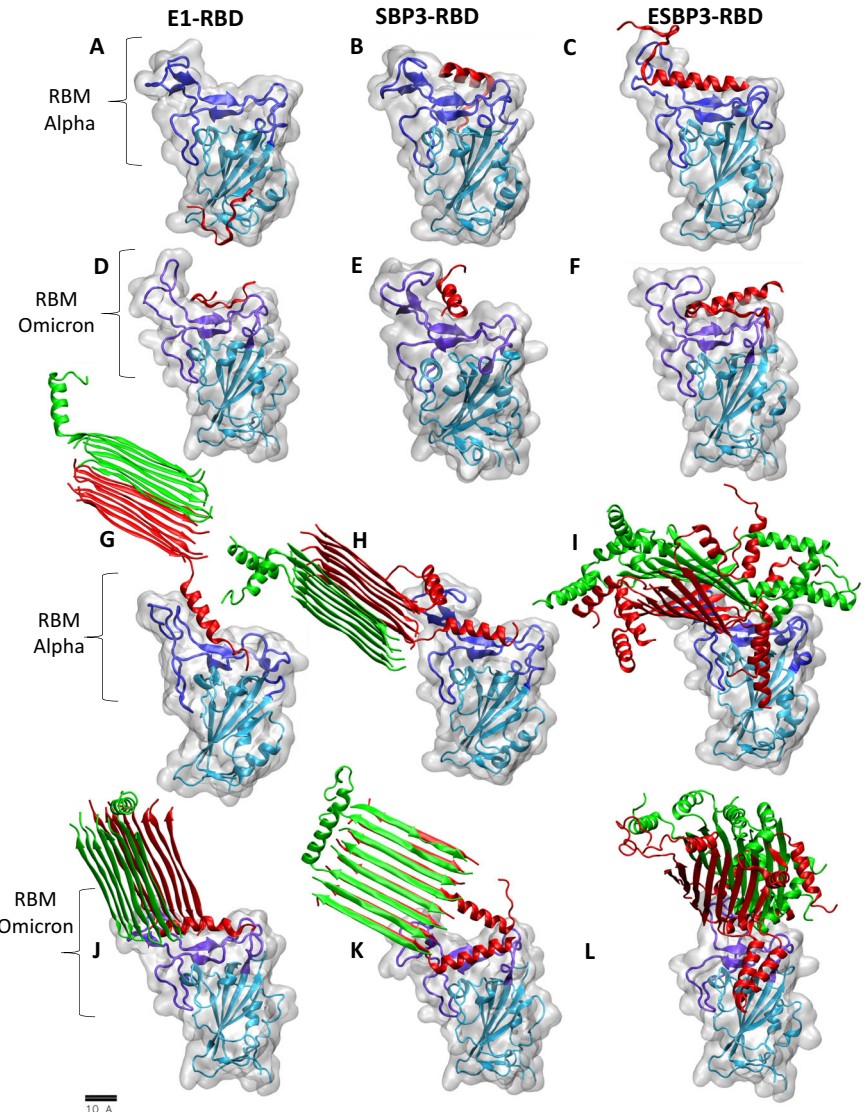

**Fig. 5 | Evaluation of monomer and multimer binding on variants.** MD simulations of the peptide monomers (**A**–**F**) and multimeric fibers (**G**–**L**). The peptides are represented in green and red, while the Spike RBD protein is represented in gray with a cyan backbone. The Spike RBM (residues SER438→TYR508 SER440→TYR505 for Omicron[69,95]) is represented with a purple backbone. MD simulations of binding the E1 self-assembling domain against COVID Variant B.1.1.7 (**A**) showing limited specificity against the RBM compared with SBP3 (**B**) and ESBP3 (**C**). Conversely, E1 remained in the binding pocket against the B.1.1.529 variant, likely as a result of the higher concentration of positive surface charge in the binding pocket of the B.1.1.529 variant (**D**). SBP3 bound the B.1.1.529 variant more effectively than the original variant (**E**), but ESBP3 showed improved binding to the Omicron variant

(**F**). To evaluate how the targeting domain of ESBP3 is affected by the presence of other ESBP3 monomers in the fiber, we performed MD simulations of 1:7 ESBP3:E1 against the Alpha variant (**G**), 1:3 ESBP3:E1 against the Alpha variant (**H**), all ESBP3 against the Alpha variant (**I**), 1:7 ESBP3:E1 against the Omicron variant (**J**), 1:3 ESBP3:E1 against the Omicron variant (**K**), and all ESBP3 against the Omicron variant (**L**). We observed a relationship between the concentration of ESBP3 and multivalency and self-interference between the bioactive domains of the monomers. As the concentrations of ESBP3 increase, there are increasing clashes and self-ligation between the bioactive domains of ESBP3 monomers (**K**), disorder in the β-sheet (**I**, **L**), and non-specific interactions with the receptor (**I**, **L**).

(SALEEQLKTFLDKFMHELEDLLYQLAL) was derived by Karoyan et al.[23]. The SWISS-MODEL Expasy web server[65–68] was used to create structural models of peptides presented in prior literature and which were mutated from ACE2 α1 helix residues 19–45 using homology-based modeling of the SARS-CoV-2 ACE2-RBD complex PDB ID: 6M0J[69]. We used the HADDOCK 2.4 web server for molecular docking runs between each peptide and Spike RBD using default docking parameters, and six complexes from the cluster were selected based on predicted energy of interaction for further analysis and comparison to the ACE2-RBD complex[51,70]. A predicted binding energy and dissociation constant ($K_D$) were generated under standard conditions for each complex using the PRODIGY web server (https://bianca.science.uu.nl/prodigy/)[71], and Gibbs Free Energy values were predicted

using the BalaS BUDE Alanine Scanning web server (https://pragmaticproteindesign.bio.ed.ac.uk/balas/)[52]. The latter web server also provided insights into the effect of a synonymous mutation at each amino acid (R→Ala) in the peptides on binding, enabling the identification of crucial residues that made a significant contribution based on their differential Gibbs Free Energy (ΔΔG) values. Residues with a ΔΔG higher than 0.718 kcal/mol were considered most impactful, and residues with a ΔΔG higher than 0.239 kcal/mol were considered essential[72].

## Development of SBP3

The overall design strategy was guided by the SARS-CoV-2 Spike-RBD bound ACE2 complex (PDB ID: 6M17 and 6M0J)[10,73]. SBP3

(QYKTYIDKNNHYAEDERYK) was developed by comparing the BAlaS BUDE outputs for SBP1 and SBP2 peptides. The residues that had large, positive ΔΔG values were scrutinized and considered for the development of the novel constructs. A new peptide sequence was created from a combination of these significant residues, which included residues conserved from the original ACE2 α1 helix and residues from the tested peptides. Two amino acids had comparable ΔΔG values at three positions in the peptide (10, 15, and 17), necessitating the creation of $2^3 = 8$ variant sequences.

Each sequence was validated using the protocol outlined in the "Evaluation of SBP1 and SBP2" section above. Secondary analysis was performed using the BeAtMuSiC v1.0 web server (http://babylone.ulb.ac.be/beatmusic/)[74]. In addition to assessing the effects of an R→Ala mutation, the BeAtMuSiC web server assesses the effects of every possible synonymous mutation at each amino acid residue. The ΔΔG values obtained from this analysis were used to create a variant peptide, but this peptide performed poorly in the evaluation and thus omitted from this work[75].

## Development of ESBP1, KSBP1, ESBP2, and ESBP3

The self-assembled analogs were developed by conjugating the SBP mimics (SBP1, SBP2, SBP3) into previously published sequences E1 (ESLSLSLSLSLSLE)[24,48] or K1 (KSLSLSLSLSLSLK)[28,76] using a glycine spacer. The existing self-assembling peptide K-(SL)$_6$-K or E-(SL)$_6$-E backbone was conjugated to an SBP mimic with a glycine spacer yielding X-(SL)$_6$-X-SBPy—where X is E or K and y is SBP1/2/3, onto which a peptide could be added at the C-terminus. All peptides were dissolved in pharmaceutical-grade saline (0.9% NaCl).

## Generation of receptor-binding domains of α and o variants

Spike receptor-binding domains from SARS-CoV-2 variants (alpha, α, GISAID Access ID: EPI_ISL_674612hCoV-19/England/MILK-B8DE60/2020; omicron, GISAID Access ID: EPI_ISL_6795848hCoV-19/South Africa/CERI-KRISP-K032367/2021 o) were modeled using SWISS-MODEL (https://swissmodel.expasy.org/interactive)[65] by providing the variant sequence of the whole Spike trimer and using the top SWISS-MODEL generated structure (assessed with GMQE score) as the homology template. The trimer structure was further truncated to amino acids 333 to 526 to create the RBD. The original RBD on the peptide-RBD complex was carefully replaced with variant RBDs after alignment. The peptide sequences were generated using the fab command for amino acids on Pymol. The peptide-RBD complexes were relaxed on Rosetta with a flexible peptide backbone for readjustment on the new complexes[77]. Successful complex alignments and relaxations were verified with a record of binding energies on BUDE Alanine Scanning[52].

## Generation of the multimers for fiber analysis

The multimeric fibers were generated on Pymol by arranging the fibers 5 Å apart in the fiber-long dimension. The E1 fibers were arranged to face parallel or antiparallel. The ESBP3 fibers were placed antiparallel, consistent with the hypothesized assembly for multidomain nanofibers. E1:ESBP3 fibers were assembled in a 3:1 E1:ESBP3 molar ratio. All fibers for MD analysis were 16-mer in length and underwent a flexible backbone relax protocol in Rosetta prior to simulation[77].

## Molecular dynamics simulations

Atomistic MD simulations were performed using NAMD3[78,79] with the CHARMM36 protein force field and TIP3P water[80]. The simulations were performed in a physiological solution with 0.15 M NaCl. The cutoffs of van der Waals (vdW) and Coulombic coupling were 10 Å. The particle mesh Ewald (PME)[81] method was used for the evaluation of long-range Coulombic interactions and to increase the efficiency of the simulations. The simulations were performed in the NPT ensemble ($P = 1$ bar and $T = 310$ K). The systems were simulated for 400 ns using the Langevin dynamics with a damping constant of 1 ps$^{-1}$ and a time

step of 2 fs, and repeated 3 times to ensure consistency for initial configuration-dependent structures (Figs. 5H, I, K, L, SI Figs. 6G, H). The monomer on receptor simulations and the 1–7 ESBP3-E1 complexes (low local SBP3 density) were simulated for 800 ns to ensure sufficient sampling of the energy space.

The trajectories were assessed for energy vs. time to determine areas of a localized flat energy minima for analysis, the details of which are present on SI Figs. 5 and 6. These MM-GBSA energies were calculated from 360–400 ns for the E1 fibers (Fig. 4A–C). The 3 Å ASA was calculated from 360–400 ns for the E1-ESBP3 combinations.

The Coulombic and van der Waals contributions to the coupling energies of interacting residues in different molecules were calculated by the NAMD energy plugin. The electrostatic contributions were obtained from,

$$U_{elec} = \sum_{i=1}^{n} \sum_{j>i}^{n} \frac{1}{4\pi\varepsilon_0\varepsilon_r} \frac{q_i q_j}{|\vec{r}_i - \vec{r}_j|} \tag{1}$$

where $q_i$ and $q_j$ are charges in the molecules, $\vec{r}_i$ and $\vec{r}_j$ their respective positions, and $\varepsilon_0$ is the permittivity of vacuum and $\varepsilon_r$ is the dielectric constant of water.

The van der Waals and close-distance atomic repulsion contributions were calculated from the Lennard-Jones (LJ) 6–12 potential energy,

$$U_{LJ} = \sum_{i=1}^{n} \sum_{j>i}^{n} \varepsilon_{ij} \left[ \left( \frac{\sigma_{ij}}{r_{ij}} \right)^{12} - \left( \frac{\sigma_{ij}}{r_{ij}} \right)^{6} \right] \tag{2}$$

Here, $\varepsilon_{ij}$ is the maximum stabilization energy between $i$th and $j$th atoms, where $\sigma_{ij}$ is the distance between the atoms at the minimum potential energy, and $r_{ij}$ is the actual distance between the two atoms. Lorentz–Berthelot rules were used to calculate the LJ parameters between different atom types[47]. Molecular mechanics with generalized Born and surface area solvation (MM-GBSA) calculations have been performed to find the free energies of binding between the self-assembling domains[82–84]. Separate MM-GBSA calculations were done for three components: a self-assembling monomer, a self-assembled 16-mer fiber with one monomer subtracted, and the whole complex. The approximate free energies of binding for the studied complexes were calculated as

$$\Delta G_{MMGB-SA} = G_{TOT(E1-fiber)} - G_{TOT(E1)} - G_{TOT(fiber)} \tag{3}$$

Where the MM-GBSA free energy for each of these molecules or their complexes were calculated from,

$$G_{TOT} = E_{MM} - G_{solv-p} - G_{solv-np} - TS \tag{4}$$

Here, $E_{MM}$ are the internal molecular energy, $G_{solv-p}$ is the polar contribution and $G_{solv-np}$ is the non-polar contribution to the molecular solvation energy, and $TS$ is the entropic contribution to the free energy. NAMD3 was used to calculate the first 3 terms of $G_{TOT}$ in implicit solvent with a dielectric constant of water of $\varepsilon_r = 78.4$. The $E_{MM}$ term is calculated by summing electrostatic energy, van der Waals energy, and internal energy (bond, angle, and torsional energy contributions). The $G_{solv-p}$ term is obtained by using the generalized Born (GB) model whereas the $G_{solv-np}$ term for each system configuration was calculated using a solvent-ASA, which was evaluated along the entire simulation trajectory, where a radius of 1.4 Å was used of the mimetic regions using a surface tension of $\gamma = 0.00542$ kcal mol$^{-1}$ Å$^{-2}$. The $TS$ term was neglected, since the entropic contribution differences nearly cancel out when we consider protein–protein binding interactions[85,86].

## Solid-phase peptide synthesis

Peptides were prepared on a LibertyBlue solid phase peptide synthesizer (CEM, Matthews, NC) using standard Fmoc chemistry, (1:4:4:6 resin:amino acid:1-[bis(dimethylamino)methylene]-1H-1,2,3-triazolo[4,5- b] pyridinium 3-oxid hexafluorophosphate:diisopropylamine, 0.1 mM scale) and all peptides were N-terminally acetylated (achieved with a 1:3 ratio of acetic anhydride to diisopropylethylamine in dichloromethane) and C-terminally amidated (low loading Rink amide resin, 0.36 mmol/gram). After synthesis, the crude peptides were cleaved with 0.25 mL each of MilliQ water, triisopropylsilane, and 3,6-dioxa-1,8-octanedithiol (DoDT), and 9.25 mL trifluoroacetic acid for 30 min at 37 °C[32,33]. Post cleavage, the crude peptides were filtered with a fritted column and triturated with cold ether. The ether and peptide mixtures were vortexed and centrifuged, the ether was then decanted, and the crude peptides were left to dry overnight. The following day, the crude peptides were dissolved at ~1 mg/mL in MilliQ water (pH ~7.0). These peptide solutions were dialyzed (Spectra/Por S/P 7 RC dialysis tubing, 2 kDa MWCO) against deionized water for 3 days (water changed 3× daily, 1:1000 peptide to reservoir volume). The peptides were then frozen and lyophilized to obtain the final peptides. The expected molecular weights of the peptides were confirmed with LC/MS. Peptides were reconstituted at room temperature at 10 mg/mL (or stated molarities) in saline with dissolution in <1 minute, and stored at 4 C till used[49,76,87]. Mixtures of peptides were formulated at equimolar (or stated) concentrations by mixing solubilized formulations, brief vortexing, and sonication prior to use.

## LC/MS and stability testing

Chromatographic analyses were performed on an analytical high-pressure liquid chromatography system (HPLC) with MassLynx software (Version 4.1). Runs were carried out on a Zorbax C3 column (5 μm, 150 mm × 4.6 mm) with a 1.0 mL/min flow rate. The mobile phase A was 0.1% trifluoroacetic acid in MilliQ water and mobile phase B was 0.1% trifluoroacetic acid in acetonitrile. Gradients started with mobile phase A at 75% and ran at a 1% ramp to 65% mobile phase B. Samples were diluted using 3:7 ACN:MilliQ water to a concentration of 1.0 mg/mL and then passed through a 0.22 μm syringe filter before injecting 30 μL of the sample. The column and autosampler were maintained at 40 °C and 25 °C, respectively. A Waters 2487 UV detector monitored 215 nm, and mass spectra were collected from 50 to 3000 AMU. The capillary voltage was 3.00 kV, cone voltage was 30.00 V, extractor voltage was 3.00 V, RF lens voltage was 0.1 V, source temperature was 100 °C, cone gas flow was 50.0 L/h, and desolvation gas flow was 500 L/h[30,88] for the electrospray source.

For stability testing, ESBP3 was incubated both as lyophilized powders (~5 mg per vial) and in formulation (~100 μL /vial) at −20 °C (lyophilized only), 4 °C, 25 °C and 37 °C for 12 weeks. The stability of peptide samples was then measured using HPLC[87].

## Fourier transform infrared spectroscopy (FTIR)

Fourier Transform Infrared Spectroscopy experiments were performed on a PerkinElmer IR 100 spectrophotometer (MA, USA) in attenuated total reflectance mode. Samples were prepared with a concentration of 0.1–0.01 mg/mL in saline. A saline background was obtained for each reading, and sample spectra were collected between 400 and 4000 cm⁻¹. The spectral region of 1400–1700 cm⁻¹ was displayed to highlight the amide I and amide II regions[28,76].

## Circular dichroism (CD)

To determine peptide secondary structure, circular dichroism (CD) experiments were performed using a Jasco J-810 spectropolarimeter (Oklahoma City, OK, USA). The peptide (s) were dissolved at 0.1 to 0.01 mg/mL in saline, and 400 μl of the sample was pipetted into a 10 mm cuvette at room temperature. Data was recorded from 190 to 260 nm[24,28,48,76].

## Mechanical testing (rheometry)

The thixotropic nature of the hydrogel was evaluated with a Texas Instruments (ARES G2) oscillatory rheometer. The hydrogel was prepared at 1w%, or 10 mg/mL in 1× PBS, and 40 μL was pipetted in between the rheometer bottom plate and an 8 mm geometry. A shear viscosity test was performed, plotting shear rate (1/s) and viscosity (1/Pa) on the x and y axis, respectively[89].

## Atomic force microscopy

Atomic force microscopy (AFM) characterized fibrillation of β-sheet-forming peptides. Samples (10 mg/mL) were prepared first in saline, then diluted with MilliQ water to 1 and 0.1 mg/mL and pipetted onto freshly cleaved mica. Peptide solutions were pipetted 3× (10 μL aliquots) onto a rapidly spinning centrifugal disk to evenly spread the peptide solution. All samples were imaged in ScanAsyst mode (Bruker Dimension Icon instrument, AZ, USA) with sharpened silicon (0.2–0.8 N/m, Al reflective coating) AFM tips[30,48,76,88].

## Scanning electron microscopy

Scanning electron microscopy (SEM) characterized the supramolecular structure of formulations. Carbon tape was used to adhere lyophilized peptide and then sputter coated with Au/Pd (8 nm thickness, EMS 150 TES sputter coater, Quorum, East Sussex, UK) and imaged with a JSM-7900 (Jeol, Peabody, MA) scanning electron microscope (5.0 kV accelerating voltage, 10 mm working distance)[30,48,76,88].

## Transmission electron microscopy

Negative stain transmission electron microscopy (TEM) was performed using standard procedures. Briefly, Holey carbon-coated, glow-discharged 400-mesh electron microscopy grids (Electron Microscopy Sciences, Hatfield, PA) were loaded with (i) ESBP3 only (10 μM), and (ii) ESBP3 pre-incubated with viral RBD (10:1), stained with 1% (v/v) ammonium molybdate (in $H_2O$, pH ~5, 2 minutes), washed with water and air-dried before being imaged on a JEOL 2200FS (200 kV) electron microscope.

## ITC

Calorimetric experiments of binding between (i) ESBP3 and SARS-CoV-2 Spike-RBD, and (ii) self-assembly of ESBP3 peptide only were carried out using MicroCal Auto-iTC$_{200}$ instrument (GE Healthcare). ESBP3 and RBD were suspended in the ITC buffer (1× PBS). RBD (5 μM in the sample cell) was titrated against ESBP3 (100 μM in the syringe), and ESBP3 (1 mM in the sample cell) was titrated against ESBP3 (4 mM in the syringe) with a constant stirring speed of 907 rpm. 20 injection titrations were carried out at room temperature. The reference power and injection volumes were kept 10 μcal/sec and 2 μL, respectively. The experiments were reproduced to confirm the findings. Each titration was fitted into a one-site binding model by using the Origin software provided with the instrument.

## Solid-state nuclear magnetic resonance (NMR) spectroscopy

To experimentally validate E1:ESBP3 peptide mixtures, solid-state NMR experiments were pursued. ¹³C solid-state NMR measurements were conducted using cross-polarization magic angle spinning (¹H-¹³C CPMAS) to evaluate sample structural order and structural differences between samples. The peptides were not isotopically enriched, so the ¹³C signal was from the 1% natural abundance for ¹³C. Furthermore, signals detected via CPMAS measurements resulting from the rigid regions of the self-assembly (the core amyloid structure—owing to E1), as dynamic solvent-accessible regions are not expected to exhibit measurable cross-polarization effects[90].

After peptide dissolution at 10 mg/mL, the samples were concentrated by ultracentrifugation for 30 minutes in Bruker 3.22 mm NMR rotors at 4 °C and 150,000 RCF in Ultra-clear tubes of SW-41 Ti

swinging bucket rotor fitted onto a Beckman Optima XPN-100 centrifuge. A 11.75 T magnet Bruker Avance III HD 500 spectrometer (500 MHz, $^1$H NMR Larmor frequency) equipped with a room temperature 3.2 mm Bruker Low-E $^1$H/$^{13}$C/$^{15}$N NMR probe was used for measurements. $^{13}$C chemical shift referencing was conducted using adamantane referenced to tetramethylsilane. 1D $^{13}$C cross-polarization magic angle spinning (CPMAS) spectra were collected using the standard Bruker *cp* pulse sequence. A 10 kHz magic angle spinning speed was used for all samples. The channels frequency for the optimization of the Hartmann-Hahn condition for $^1$H and $^{13}$C are 500.133 MHz and 125.771 MHz, respectively. A recycle delay time of 4 seconds (d1) and dwell time of 5 μs (DW) were used. Acquisition time was set to 10.24 ms. Signals were averaged over ~12 hours at room temperature. Data were processed using TopSpin v3.6.2, and custom code developed using Wolfram Mathematica 13.

### Solution nuclear magnetic resonance (NMR) spectroscopy
Peptide samples were dissolved in 99.9% deuterium oxide ($D_2O$) at a concentration of 1 mg/mL prior to solution NMR measurements recorded at 25 °C on an 11.75 T Bruker Avance III HD 500 spectrometer (500 MHz, $^1$H NMR Larmor frequency) equipped with a liquid nitrogen-cooled Prodigy cryoprobe. 1D $^1$H solution NMR spectra were collected using standard Bruker pulse sequence *zgesgp* for solvent suppression using excitation sculpting with number of scans set to 100, recycle delay set to 2 seconds (d1), acquisition time set to 3.28 seconds, and dwell time set to 100 μs (DW). For 1D $^1$H Carr-Purcell-Meiboom-Gill (CPMG) solution NMR experiments were collected using standard Bruker pulse sequence *cpmg_esgp2d* with a number of scans set to 64, recycle delay set to 5 seconds (d1), dwell time set to 62 seconds (DW), acquisition time set to 1.02 seconds, CPMG time (d31) set to 0.002 seconds, and vclist values of 5, 25, 50, and 800 (corresponding to shorter to longer CPMG/$T_2$ relaxation filter values). Measurements were conducted at room temperature. Data were processed using TopSpin v3.5 and visualized in Wolfram Mathematica 13.

### Cytocompatibility
Cytocompatibility of the peptides was evaluated at 1, 0.1, and 0.01 mM with A549 cells (ATCC CCL-185) cultured in complete media consisting of 90% F-12K and 10% FBS. The A549 cells were utilized after their first passage and seeded at 10,000 cells/well in a 96-well plate for 24 hours. The conditions were prepared in serum-free A549 media and introduced into the wells for 6 hours, after which the condition media were aspirated, the wells were washed once with PBS, and 10 μL of CCK8 was added to 100 μL of PBS[32]. The plate was incubated for 1 hour and read on a Tecan M200 Infinite plate reader at an absorbance of 450 nm with a reference wavelength of 650 nm. The results were analyzed and normalized to the serum-free media control[28,32,76].

### In vitro efficacy testing of peptides against SARS-CoV-2 Spike pseudoviruses
The SARS-CoV-2 Spike pseudotyped viruses were prepared using HIV-1 plasmid with luciferase reporter, pNL4-3-Luc, and pcDNA3.1(+) containing SARS-CoV-2 Spike gene (Wuhan-Hu-1 (Acc. No. QHD43416)) as described[91]. The plasmids were co-transfected into HEK-293T (ATCC CRL-1573) cells using Lipofectamine™ 3000 Transfection Reagent (Cat. No. L3000001). The transfected cells were supplemented with serum-free DMEM, and the serum-free spent media of transfected cells containing pseudovirus were harvested at 48 hours post-transfection. The virus was centrifuged at $600 \times g$ for 5 minutes at 4 °C to remove the cell debris filtered through 0.4 μm filter, and stored at −80 °C for further use.

For the titration of pseudoviruses, HEK-293T-ACE2 cells were seeded at $5 \times 10^4$ cells/well onto the 48-well cell culture plate. At 18–24 hours of seeding, the supernatant was removed, and cells were infected with 100 μL of pseudovirus at 37 °C for 1 hour, and then the infected cells were replenished with 400 μL of DMEM containing 10% FBS. At 72 hours post-infection, the media were removed, and 60 μL of luciferase lysis buffer was added to the cells and incubated at room temperature for 30 minutes. Then 40 μL of lysate was transferred to Microlite 2+ flat bottom white-well luminescence plate, and 25 μL of luciferase reagent was added to the lysate, and luminescence (RLU) was measured immediately.

HEK-293T-ACE2 cells were seeded in the 48-well plate. Next day, 500 μL of serum-free SARS-CoV-2 Spike pseudoviruses with known infectivity (RLUs) were incubated with 0.01 μM, 0.1 μM, 1 μM, 10 μM peptides at 37 °C for 1 hour while keeping untreated pseudovirus control. Then pseudovirus/peptide complex or pseudovirus at 100 μL were added to the HEK-293T-ACE2 cells and incubated at 37 °C for 1 hour and replenished with 400 μL of DMEM supplemented with 10% FBS. At 72 hours post-infection, the cells were lysed, and luciferase was measured. The RLUs for virus control and peptide treatments were plotted against the concentrations of each peptide. The pseudovirus entry was quantitated as the luciferase expression in the infected cells.

### In vitro efficacy testing of peptides against SARS-CoV-2
Propagation and titration of SARS-CoV-2 was performed as we described previously[92]. Briefly, Vero E6 (ATCC CRL-1586) cell monolayer was infected with SARS-CoV-2 at MOI of ~0.1, briefly, the spent media (DMEM containing 10% FBS) from Vero E6 monolayer was decanted and washed with PBS (pH ~7.2). The monolayer was infected at 37 °C for 1 hour and then replenished with DMEM containing 2% FBS. The cell culture supernatant containing the virus was harvested at 72 hours post infection. The virus aliquots were stored at −80 °C[92].

Virus infectivity was quantitated by plaque assay using Vero E6 cells, in brief, $4 \times 10^5$ Vero cells/well were seeded on the six-well plate using DMEM containing 10% FBS. After 18–24 hours the cells were washed with PBS (pH ~7.2) and infected using 400 μL of 10-fold dilutions of virus at 37 °C for 1 hour. Then the virus inoculum was removed and 4 mL/well of agarose-DMEM overlay (50 mL of overlay contains 25 mL 1.6% agarose and 25 mL 2× DMEM with 8% FBS) was added to the wells. The plates were incubated at 37 °C with 5% $CO_2$ for 3 days. The cells were fixed with 10% buffered formalin, agar plugs were removed, stained with 0.2% crystal violet (in 20% ethanol), and plaques were counted[92].

About 30–40 plaque-forming units of SARS-CoV-2 in serum-free DMEM were incubated with the peptides at 0.01 μM, 0.1 μM, 1 μM, 10 μM peptides for 1 hour at 37 °C (untreated virus control was maintained in each plate). Then, the Vero E6 cell monolayer was washed with PBS and incubated with virus-peptide complex or virus and incubated at 37 °C for 1 hour. The inoculum was removed and 4 mL agarose-DMEM overlay was added to each well. The plates were incubated for 3 days at 37 °C with 5% $CO_2$. The plates were fixed with 10% buffered formalin, agar plugs were removed, stained with 0.2% crystal violet (in 20% ethanol), and plaques were counted[93]. $IC_{50}$ was calculated using a four-parameter logistic regression model (Quest GraphTM CC50 Calculator, AAT Bioquest (https://www.aatbio.com/tools/ic50-calculator))[94].

### In vivo assays
All animals were treated in accordance with NJIT-Rutgers Newark Institutional Animal Care and Use Committee (IACUC) policies and AALAC, and AWA guidelines. Housing conditions for the mice/rats, have 12 hr dark/light cycle, ambient temperature, and humidity. All animals were purchased from Charles River Laboratories and housed at local facilities for a minimum of 1 week before studies were commenced.

### Biocompatibility−rodent sub-Q analysis
The peptide hydrogel was prepared at 10 mg/mL in sterile saline and aspirated in sterile 1 mL syringes, with careful attention to avoid air bubbles. Female Wistar rats (225–250 g, 8–12 weeks old) were used for dorsal subcutaneous implantation. The rats were anesthetized using

2.5% isoflurane and 2% oxygen, after which the back of the rat's hair was shaved and sterilized with 3 alternating applications of isopropanol and betadine. 200 μL boluses of the peptide hydrogel were injected with a 25 G needle at $n = 4$ for a time point of 7 days. After the time point, the animals were sacrificed, and the implant regions were collected and fixed overnight in 10% buffered formalin. The Rutgers Cancer Institute of New Jersey histology core processed the paraffin-embedded samples for histological evaluation after staining with hematoxylin and eosin and Masson's Trichrome. Samples were also fixed in glutaraldehyde and formaldehyde, processed to eponate blocks, and sectioned with an ultramicrotome for electron microscopy imaging (above)[30,76,87].

### Trafficking—rodent IV repeat dose and PK analysis

Female Wistar rats (225–250 g, 8–12 weeks old) were observed and acclimatized for 2 weeks prior to running the studies. They were housed in a controlled environment with temperature at -37 °C and 40–70% humidity. They were kept in an alternating 12-hour light/dark cycle, and food, and water was provided as pellets ad libitum. Female Wistar Rats were used for the pharmacokinetics and the drug clearance analysis from the blood. The two hydrogels (ESBP2 and ESBP3) at 10 mg/ml concentration were injected into the rats via the lateral tail vein. For subcutaneous implants, the substance was injected between the shoulder blades and above the lumbar vertebrae, 200 μL each, using a 25-gauge needle. The hydrogel bolus was prepared with 0.9% saline and was injected within 1–2 minutes. The volume injected was approximately 1% of the total body weight of the rat, which was calculated prior to the dosing. The initial blood draw was performed for baseline, and the subsequent blood collection was performed from the lateral vein using 25 g butterfly needles.

### Acute PK

Rat blood samples were collected at 0, 5, 15, 30 minutes, and 1, 2, 4, 8, and 24 hours post-injection timepoints for Acute PK and were aliquoted out into heparinized tubes to prevent coagulation and stored on top of ice packs. Blood collected in K2EDTA anticoagulant tubes was processed to plasma frozen via centrifugation at 13000 RCF for 10 minutes at 4 °C and stored at −80 °C till analysis using HPLC. The body weight was recorded every day prior to the blood collection.

### Chronic PK

Female Wistar rats (225–250 g, 8–12 weeks old) were weighed and repeatedly dosed (1% by weight) IV with ESBP2 or ESBP3 daily at 10 mg/mL (~3 mM). Rat blood samples were collected daily prior to injection via tail vein. Post 10 days of treatment, animals were allowed to recover for 10 days and then sacrificed. Plasma was stored and analyzed as above. The body weight was recorded every day prior to the blood collection.

### Tolerability—rodent IN repeat dose

Female C57BL/6 mice (15–20 g, 8–12 weeks old) were used for intranasal dosing of the peptide. As mentioned previously, 25 μl of hydrogels were injected into each of the nasal cavities twice daily at an interval of 8 hours for 10 days consecutively. The organs were harvested at the final time point, and sectioning was performed for histological evaluation. The plasma was also collected for quantifying peptide clearance in the bloodstream of the animal model.

### Reporting summary

Further information on research design is available in the Nature Portfolio Reporting Summary linked to this article.

## Data availability

The authors declare that the data supporting the findings of this study are available within the paper and its Supplementary Information files.

Input files for all MD simulations are available: https://github.com/jbdoddo/COVID_PEPTIDE_MS. Should any raw data files be needed in another format, they are available from the corresponding author upon reasonable request. Source data are provided in this paper within the "Source Data" file. PDB entries relevant to this study: PDB ID: 6M17 (https://doi.org/10.2210/pdb6M17/pdb) and 6M0J (https://doi.org/10.2210/pdb6M0J/pdb) Source data are provided with this paper.

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

## Acknowledgements

We thank the support from Mario Borgnia, Kedar Sharma, and Lucas Dillard of the Genome Integrity and Structural Biology Laboratory, National Institute of Environmental Health Sciences, NIH, DHHS, for their assistance in cryo-EM and reconstructions. V.K. acknowledges support from the NIH NEI R15 EY029504, NIH NIAMS R21AR079708; and the Undergraduate Research and Innovation program at NJIT. VK and CH acknowledge funding from NSF IIP 2032392. A.S.R., J.L., and A.K.P. acknowledge support from the NIH RF1AG073434-01A1. P.K. acknowledges support from NSF DMR 2212123.

## Author contributions

V.K. conceived of the idea and supervised the research. V.K., P.K., B.V.V.P., A.P., and C.H. designed the experiments. J.D., A.R., and Z.S. performed peptide synthesis, characterization, and in vivo biocompatibility studies. J.D., R.J., F.C. A.A. V.P. A.A.-J. performed computational designs and simulations. S.R., A.K., and R.K. performed in vitro pseudo- and live virus assays. D.K., S.K., and B.Z. performed negatively stained EM and ITC measurements. A.S.R., J.L., and A.M. performed NMR experiments. A.L. performed histopathological analyses of tissue. All authors analyzed the data, participated in the scientific discussion, and contributed to the writing of the manuscript.

## Competing interests

V.K. and N.J.I.T. have filed a non-provisional patent on this and related technologies; V.K. and C.H. have equity interests in startups related to the translation of this platform technology. The remaining authors declare no competing interests.
