## [Peer Review File · Nature Communications]

Reviewers' Comments:

Reviewer #1:

Remarks to the Author:

This manuscript describes the development of a set of self-assembled peptides designed to inhibit entry of SARS-CoV-2 into mammalian cells by binding to and obstructing the receptor binding domain (RBD) on Spike proteins. Starting with peptide sequences selected from the human ACE2 receptor, the manuscript describes computational optimization of the binding sequences to improve binding to Spike proteins *in silico*. These binding sequences were synthesized in tandem with an established self-assembly domain and tested experimentally for their ability to prevent viral entry in a cell line from non-human primates, as measured by reductions in plaque formation. An extensive amount of detailed characterization of the peptides was performed, as well as safety testing after *in vivo* injections of the self-assembled constructs in *vivo* in mice; much of this characterization was relegated to the supporting material. Finally, molecular dynamics simulations were performed to help explain the differences in *in vitro* efficacy observed between the different constructs.

In principle, development of a well-characterized neutralizing peptide assembly to help combat SARS-CoV-2 infection, or other viruses in the future, by inhibiting viral entry is a potentially impactful strategy and could be an important advance for the field of anti-viral development. However, currently the manuscript is missing some critical context, and the data is challenging to interpret in some places. More important, near the end a piece of data appears to call into question the central premise that the self-assembled peptides are specifically targeting the RBD (see point 1 below), but this observation was given only brief mention with little explanation. Therefore, major revisions are required to ensure that the conclusions of the manuscript are fully supported and that the impact of the work is clear. Specific comments are given below:

Major

1. Is it a surprise that 100 nM of ESBP3 was no more effective than 100 nM of the E1 carrier alone (Figure 4J)? To me, this is a non-intuitive result and merits substantial discussion in the Results, considering that in Figure 3G, there was a clear dose-response to ESBP3. These data might suggest that most or all of the observed dose-response to ESBP3 in Figure 3G, and even the response to ESBP1 in Figure 2, could actually be a non-specific response to the E1 carrier. In fact, a statement in the Conclusions section appears to support this interpretation: "Interestingly, E1 alone showed broad antiviral potential principally due to its net charge interactions with viral protein coats, as reported for several similar poly-anionic scaffolds for SARS-CoV-2 treatment". Currently, the manuscript is framed around the specificity of the RBD-binding peptides, but based on these data it is possible that in fact those may have only a minor role compared to the scaffold itself. This issue should be discussed in the manuscript, and ideally additional data would be shown, such a measurement of the dose-response of the E1 scaffold on plaque formation.

2. One of the central means of testing the efficacy of the peptides and self-assemblies was an *in vitro* plaque formation assay. According to the Methods, the peptides and self-assemblies were allowed to bind to the virus first in serum-free medium before testing infectivity towards the cells. However, presumably for use as a therapeutic, the peptides would need to be injected *in vivo* in infected patients, where they would need to bind tightly even in the presence of abundant albumin and other proteins. It would be useful to test binding with and without serum present, and/or to test plaque formation with and without serum present during the binding step. If this is not feasible currently due to practical constraints, then this limitation of the study should be discussed explicitly in the text.

3. Figure 5 appears to illustrate results of MD simulations in terms of the position of the peptide upon binding to the receptor binding domain, but the text and caption refer to these data as though they show "affinity". Traditionally affinity means the K_D of binding, a quantifiable metric, but currently I do not see such numerical data in the figure. This data should be provided, or the term "affinity" should be removed from the text accompanying this figure.

4. In terms of positioning this work in the context of the field, there were two areas that should be given additional attention in the Introduction. First, the concept of using self-assembled peptides (SAPs) for multivalent presentation of therapeutics or vaccine antigens is well grounded in prior

literature on topics such as self-assembled peptide-based vaccines, and self-assembled peptide-based multivalent drug delivery. More of this literature needs to be highlighted in the introduction, which currently discusses and cites prior use of SAPs primarily in the context of physiochemical properties such as increased binding affinity and multivalency. See, for example, extensive prior work in this area by Joel Collier et al (one such paper is cited but not for its application to vaccination), Jai Rudra et al, and many others that can be found by searching for self-assembled peptide based vaccines or self-assembled peptide based therapeutics. The current work is different from a vaccine in that it is a neutralizing reagent, but these other works serve as a key foundation for feasibility of in vivo efficacy.

5. In addition, prior work on related systems using SAPs to passivate or neutralize viruses should be mentioned, as it is highly related and provides context for this new work. See, for example, materials development work by Nahhas & Webster for a passivating SAP for Omicron, which is similar in some ways to this work <https://doi.org/10.1016/j.onano.2022.100054>.

Minor

6. In the Introduction, paragraph 1, it states that the mRNA vaccines for SARS-CoV-2 "have proven to be an effective prophylactic strategy against infection of early strains". Technically, the mRNA vaccines are an effective prophylactic against severe disease and death after acute infection, but they have done little to prevent actual infection, which is one reason why the virus continues to spread and cause mild-to-moderate acute infection and more serious long term consequences. It would be best to correct this language in the introduction to avoid confusing readers.

7. It was challenging to discern what was a new result versus a reference from the literature in the first paragraph of the Results & Discussion section (discussing Figure 1). It would be useful to further clarify what part of the stated binding results and viral inhibition results are new in this work versus coming from the literature.

8. In the illustrations of Receptor Binding Domains (figures 1, 5), it would be useful to use an arrow or other indicator to point out the critical binding pocket for non-experts.

9. It is unclear how the stated IC50 value of ~1 uM was arrived at from Figure 3G, as the response did not reach saturation in the data shown. Looking at the plot, the concentration at which half of the plaque formation was inhibited appears somewhere between 1-10 µM. The calculation of the IC50 should be clarified.

10. Some of the results for the viral plaque assays are reported as dose-response data, while others are a comparison of different peptides, presumably at a fixed dose. The concentration of each peptide should be indicated in the figure or caption, e.g in Figure 2C.

11. The x-axis labels in Fig 4J are misaligned with the bars, making it difficult to know which bar is which.

12. Several of the graphs in the SI figures have extremely tiny fonts on the axes and are difficult to read. Others are low resolution (e.g. SI Fig 2D, E). These fonts should be made legible.

Reviewer #2:

Remarks to the Author:

In the manuscript entitled "Antiviral Fibrils of Self-assembled Peptides with Tunable Compositions", Dodd-o and Roy et al. designed a stabilized spike binding peptide sequence which was fused via a glycine spacer to a self-assembling peptide sequence E1, forming a self-assembling, spike binding peptide sequence (ESBP1), where the fusion of the E1 sequence nucleated self-assembly of the fibril when mixed with E1 peptide at 3:1 and 7:1 ratios. The authors demonstrate that fusion of E1 to SBP variations inactivated SARS-CoV2 pseudovirus and inhibited infection of the virion in vitro

and higher affinity SBP variations to Spike RBD enhanced ESBP inhibition of live SARS-CoV2. MD simulations showed how ESBP3 self-assembled on the alpha and omicron variants of SARS-CoV2 and NMR showed differences peak shifts in E1, ESBP3 and the E1:ESBP3 mixture, fibril self assembly.

Some suggested improvements:

Spike-binding peptides and anti-SARS-CoV2 SAPs: Why was the SAP conjugated to the N terminus of the spike binding peptides? What happens when the SAP are conjugated to the C terminus?

TEM: Some structural characterization beyond the micrograph in the TEM in Figure 3/4 should be requested, such as 2D avgs for ESBP3, E1, ESBP3+E1, and ESBP3+RBD. There is currently no higher resolution structural validation that ESBP variants+E1 mixtures actually form fibrils, despite the representative CryoEM movie in 4I. Further, how were the 1:3 and 1:7 ratios determined?

Virus inhibition: How were the relative concentrations between peptide and fibril normalized when measuring viral inhibition (4J)? It's unclear how the self-assembly of the fibril actually enhances viral inhibition relative to free non-assembled peptide, as the axes formatting is weird.

Antiviral activity of fibrils: At the moment the manuscript first focuses on comparing antiviral activity of each peptide variant without followed by modeling of fibril assembly. The authors should further emphasize the utility of antiviral fibrils vs peptides (e.g. comparison of half-life, shelf-life, serum stability, clearance, binding affinity, virus neutralization).

Binding affinity predictions: The affinity in 3C was all predicted. Is there any validation experimentally that the actual binding affinity of the SBP variants reflect the in silico modeling? Further, could the authors test whether the predicted binding orientations of SBP3 and ESBP3 peptides bind in the predicted orientations and whether the fibrils form in the predicted orientations (Fig 5)?

The authors claim larger self-assemblies can wrap the virion surface, although there is no structural data showing that the fibrils are wrapping themselves around the virion or neutralizing in some other way and are missing a citation with this claim in their introduction.

Self assembly process: Some details explaining the self-assembly process of E1 with ESBP3 could be helpful such as buffer conditions, assembly time and temperature, assembly order with virion / pseudovirus, etc.

Small improvements:

Figure 2B: The controls for E1 and K1 inhibition of SARS pseudovirus are missing.

Figure 4J: The x axis layout is confusing

Figure S1F: There is actually no comparison between ESBP2 to SBP2, as only data for ESBP2 is shown.

SUPPLIER is missing for the 400-mesh EM grids.

Reviewer #3:

Remarks to the Author:

In this study, variants of the ACE-2 helix that binds to Spike RBD were fused to a motif from serum amyloid P (SAP) to generate a fibril that displays redesigned helical moieties. Self assembly of a mixture of SAP peptide alone with conjugated SAP-helix constructs allowed for helix display without proposed crowding effects.

I was asked to comment specifically on the computational design and simulation components of this project.

(1) Regarding the design scheme, the team started with testing existing designs SBP1, SBP2 as

fusions with SAP: ESBP1 and ESPB2. HADDOCK-derived complexes of SBP2 with RBD were subject to computational Ala-scan mutagenesis to identify hot-spot residues. Other residues were truncated to allow for more room in the self-assembled construct.

This approach seems reasonable but for scientific completeness, the approach needs to be described in more detail - perhaps in the supplement. Details on the HADDOCK protocol, number of poses evaluated by ala-scanning, cutoff for hotspot/non-hotspot residues are not presented. Finally, the protocol for choosing replacement residues was not described. Whether it was rational or computational - the motivation behind the SPB3 changes should be described.

(2) Molecular dynamics is the computational tool used to interpret biophysical observations at a molecular level. Initially, it is used to differentiate between all-parallel, planar antiparallel and all-antiparallel models of the E1 assembly. This is tricky - but reasonable in the context of the current project. Structures were generated in pymol and subjected to fast-relax in rosetta. Implicit solvent, langevin dynamics simulations are performed. Nonbonding internal energetics calculated as well as protein-solvent interactions reported. It is difficult to evaluate the quality of the calculations without more details. Specifically:

- What are the criteria for convergence (energy/RMSD/geometry)? Is 400 ns enough for equilibration and production trajectory analysis? How many replicas were run for each calculation? For a system of this size, one is likely not enough. Justification and detailed presentation of simulation results should be included in the manuscript and supplement.

- Several structures of SAP are available, how do the E1 models compare? In the end, the all-antiparallel solution is the one nature chooses as well, so what is the motivation for the MD calculation?

This is overall a successful project of interest to the community if the above concerns are considered. I would just add one more general point - I could not find a hypothesis for why fusing a bioactive peptide and a self-assembling one would result in a better anti-viral. I can imagine two mechanisms - Confinement reduces the available volume, dampening dynamics and pushing the folding funnel for the helix toward a folded, helical state. In this case, similar increase in affinity could be expected for higher ratios of E1 to ESPB3. Alternatively, the increased efficacy is not affinity, but avidity. Either mechanism would have different implications for design and formulation.

Dear Reviewers,

We would like to resubmit our work “Antiviral Fibrils of Self-assembled Peptides with Tunable Compositions” by Dodd-o et. al. We have modified the manuscript with the intention of addressing all concerns when possible, and providing clarity when not. All changes in the text are highlighted. We believe that with the changes made, the work is suitable for publication in Nature Communications.

Sincerely,
Vivek Kumar

These are our answers to Reviewers:

Reviewer #1 (Remarks to the Author)

Major

1. Is it a surprise that 100 nM of ESBP3 was no more effective than 100 nM of the E1 carrier alone (Figure 4J)? To me, this is a non-intuitive result and merits substantial discussion in the Results, considering that in Figure 3G, there was a clear dose-response to ESBP3. These data might suggest that most or all of the observed dose-response to ESBP3 in Figure 3G, and even the response to ESBP1 in Figure 2, could actually be a non-specific response to the E1 carrier. In fact, a statement in the Conclusions section appears to support this interpretation: “Interestingly, E1 alone showed broad antiviral potential principally due to its net charge interactions with viral protein coats, as reported for several similar poly-anionic scaffolds for SARS-CoV-2 treatment”. Currently, the manuscript is framed around the specificity of the RBD-binding peptides, but based on these data it is possible that in fact those may have only a minor role compared to the scaffold itself. This issue should be discussed in the manuscript, and ideally additional data would be shown, such a measurement of the dose-response of the E1 scaffold on plaque formation.

Response: This is an important comment from the Reviewer that details the evolution of our design strategy – from design and evaluation of specific SARS-CoV-2 Spike binding peptides (SBP1, SBP2 and SBP3) and their conjugation with self-assembling peptides (SAP) that potentially enhance their efficacy. We have detailed the evolution of this peptide with a specific spike binding domain, and then during the study, in testing controls, noted a non-specific phenomenon with E1 – just the SAP. To determine the nature of this interaction, we compared lysine terminated peptides (such as KSBP1 vs ESBP1) - where KSBP1 showed lack of inhibition of pseudovirus compared to ESBP1, Figure 1. Complimenting this, compared to the SBP domain alone, as shown in Figure 1 - 4, when SBP1, SBP2 or SBP3 is synergized with E1, there is better presentation of the SBP3 domain, with multiple localized opportunities for SBP (in the assemble SAP) to bind and thus enhance viral inhibition. We acknowledge that this was not adequately presented in the manuscript and have thus enhanced the discussion of this. We have augmented this discussion in the Abstract and have added to the “Introduction” and “Results and Discussion” more about this non-intuitive phenomenon as suggested. Specifically, we have enhanced and focused the evolution of these peptides and the rationale for testing different formulations with the E1 domain, as suggested by the Reviewer, without being verbose/ adding too much text. For example, in the Results and Discussion, that section now reads:

“Initial attempts in peptide design involved isolation of a short Spike Binding Peptide (SBP) sequence (IEEQAKTFLDKFNHEAEDLFYQS) from ACE2 α 1¹⁷. This short sequence, termed

SBP1, was the first bioactive domain that was tested. The SAP domains were conjugated to SBP1 with 2 alternatively charged SAP terminal residues, glutamic acid (E)^{24,48} and lysine (K)^{30,31,49} (Fig. 2A). We observed a significant (~ 70 %) inhibition of pseudovirus by SBP1 from media control (Fig. 2B), as reported previously^{17,18}. E-flanked SAP termed E1 (E-SLSLSLSLSL-E) was conjugated to SBP1 (ESBP1) which inhibited pseudovirus to a similar degree, while K-flanked SAP termed K1 (K-SLSLSLSLSL-K) conjugated to SBP1 (KSBP1) had a significant decrease in inhibitory effect (Fig. 2B). A secondary B.1 lineage live virus plaque formation assay was performed with SBP1 and ESBP1 to confirm if the self-assembling domain could assist in viral inhibition. In the live virus assay, counts of infectious plaques confirmed a significant improvement in viral inhibition by ESBP1 relative to SBP1 alone (Fig. 2C).

During the testing of ESBP1 (2021), a novel spike binding domain that had a stabilized ACE2-based helix was discovered by Karoyan et al²³. The sequence was modified to include only canonical amino acids, yielding the bioactive sequence SBP2 (SALEEQLKTFLDKFMHELEDLLYQLAL) which had a strong α -helical structure (SI Fig. 1). Conjugation to the E1 self-assembling domain yielded a sequence ESBP2 (SI Fig. 1). Notably, SBP2 and ESBP2 showed excellent cytocompatibility and inhibition of live virus. ESBP2 safety was evaluated *in vivo* via 10 days repeat multi-dose IV administration of ~ 1 w% ESBP2 which showed no adverse effect in 225-250 g adult rats and ~ 4-8 hours clearance in acute 24-hour single-dose pharmacokinetic (PK) evaluation (SI Fig. 1). However, the ESBP2 peptide did not form stable β -sheets, as evidenced by an α -helical signature in CD (SI Fig. 1C) and visualized aggregates by AFM (SI Fig. 1D). Notwithstanding, this supports our hypothesis of SAP functionalization of binding domains, specifically demonstrating that SAPs with E termini can enhance Spike binding and inhibition (Figs. 2A-B).”

2. One of the central means of testing the efficacy of the peptides and self-assemblies was an *in vitro* plaque formation assay. According to the Methods, the peptides and self-assemblies were allowed to bind to the virus first in serum-free medium before testing infectivity towards the cells. However, presumably for use as a therapeutic, the peptides would need to be injected *in vivo* in infected patients, where they would need to bind tightly even in the presence of abundant albumin and other proteins. It would be useful to test binding with and without serum present, and/or to test plaque formation with and without serum present during the binding step. If this is not feasible currently due to practical constraints, than this limitation of the study should be discussed explicitly in the text.

Response: As mentioned by the Reviewer, this is an important consideration in the delivery and therapeutic translation of this technology. For this therapeutic, we envision that it might best be fit for intranasal instillation (vs IV where it may interact with serum) to prevent virions from binding to ACE-2 in the nose / turbinate as an initial mode of delivery. This would act as a “molecule mask” that shields native cells from interaction with virus particles. In order to recapitulate this *in vitro*, we incubated the peptide with pseudo- and live virus particles and then sought to determine the sustained/ diminished ability of virions to subsequently infect cells (as published by a number of other groups). Given the poor stability of short peptides in serum, and the noted binding to serum proteins (eg. albumin), we did not envision that systemic viremia could be treated with this approach of already infected cells. Notwithstanding, given the potential IV use we addressed the delivery of this therapeutic either IV or intranasally by evaluating both IV PK and intranasal

biocompatibility (Supplementary Figure 2). Further, we have enhanced the entire manuscript to address this, for example in Results and Discussion to include:

“Like ESBP2, ESBP3 showed excellent cytocompatibility with human alveolar epithelial A549 cells and dose-dependent antiviral inhibition with a calculated IC_{50} of 2.5 μ M (SI Fig. 1, Fig. 3G). ESBP3 showed rapid clearance from the circulation (< 4 hours) after IV bolus administration – characteristic of IV (peptide) therapeutics (SI Fig. 2). Daily repeated IV dosing for 10 days showed no adverse effect in rodents. Subcutaneously implanted boluses in rats and repeated daily intranasal (IN) dosing in mice showed no significant body weight changes or adverse organ morphology (SI Fig. 2). These 2 routes represented safety for potential instillation routes for prophylaxis in the nasal passageways and potential treatment IV. These biocompatibility results support safety, but may warrant future studies that probe interactions of peptides and their maintained efficacy when in contact with mucosal / plasma fluids.”

3. Figure 5 appears to illustrate results of MD simulations in terms of the position of the peptide upon binding to the receptor binding domain, but the text and caption refer to these data as though they show “affinity”. Traditionally affinity means the KD of binding, a quantifiable metric, but currently I do not see such numerical data in the figure. This data should be provided, or the term “affinity” should be removed from the text accompanying this figure.

Response: We would like to thank the Reviewer for this important comment regarding the terminology ‘affinity’ of our self-assembling peptides against receptor binding domains in the text and captions. We have removed reference to affinity and more carefully rewritten all the figure legends and text to reflect this change to remove this ambiguity and better the presentation of the results of this study.

4. In terms of positioning this work in the context of the field, there were two areas that should be given additional attention in the Introduction. First, the concept of using self-assembled peptides (SAPs) for multivalent presentation of therapeutics or vaccine antigens is well grounded in prior literature on topics such as self-assembled peptide-based vaccines, and self-assembled peptide-based multivalent drug delivery. More of this literature needs to be highlighted in the introduction, which currently discusses and cites prior use of SAPs primarily in the context of physicochemical properties such as increased binding affinity and multivalency. See, for example, extensive prior work in this area by Joel Collier et al (one such paper is cited but not for its application to vaccination), Jai Rudra et al, and many others that can be found by searching for self-assembled peptide based vaccines or self-assembled peptide based therapeutics. The current work is different from a vaccine in that it is a neutralizing reagent, but these other works serve as a key foundation for feasibility of in vivo efficacy.

5. In addition, prior work on related systems using SAPs to passivate or neutralize viruses should be mentioned, as it is highly related and provides context for this new work. See, for example, materials development work by Nahhas & Webster for a passivating SAP for Omicron, which is similar in some ways to this work <https://doi.org/10.1016/j.onano.2022.100054>

Response: We appreciate these comments on how elements of this concept have been in part presented by others, and the need to cite/ describe the rigor of prior approaches. To address them, we have expanded our discussion to better highlight this aspect, along with careful review of the current and recent literature to ensure the manuscript and discussion is more robust and up to date. We have briefly (in the interest of focus, and space) summarized and presented

seminal work on self-assembled peptide-based vaccines, and self-assembled peptide-based multivalent drug delivery in the introduction, to premise the current studies, and complimenting the existing discussion of SAPs primarily in the context of physiochemical properties such as increased binding and multivalency. As suggested by the Reviewer, we have added the citations recommended, and enhanced the entire Manuscript. For example, the revised portion of Introduction now reads:

“The early presence of a high-resolution atomistic structure of SARS-CoV-2 bound to ACE2 has facilitated rational design of novel therapeutics¹⁰ targeting the Spike receptor-binding domain (RBD) interaction with host ACE2^{2,3,6,11-16}. Based on the ACE2-RBD coupling, design strategies that employ the human ACE2 structure (binding motifs) have been identified and proposed as proteins and peptides for therapeutic development¹⁷⁻²². As promising peptide candidates display a lack in stability and live virus efficacy, molecular modeling studies were used to find site-specific amino acid mutations to optimize the helical strength, maintain low antigenicity, and high affinity for RBD^{17,18,23}. A strategy to target smaller antigenic determinants with shorter proteins/peptides may confer variant specificity, at the cost of specificity restricted to a small binding region. However, self-assembled peptide (SAP) conjugates may enhance binding^{24,25} to a target through non-covalent stabilization by multivalent²⁵⁻³⁶ supramolecular interactions³⁷. At the same time, larger self-assembled constructs can form a supramolecular assembly atop multiple RBDs at the viral surface, thereby inhibiting viral proteins from binding to cell receptors. This strategy is premised by studies that have investigated SAP-like peptides with multivalent tailorable antigen presentation³⁸ in self-adjuncting vaccines^{39,40} for upregulated targets in cancer^{41,42}, and other pathogens^{40,43}.”

Minor

6. In the Introduction, paragraph 1, it states that the mRNA vaccines for SARS-CoV-2 “have proven to be an effective prophylactic strategy against infection of early strains”. Technically, the mRNA vaccines are an effective prophylactic against severe disease and death after acute infection, but they have done little to prevent actual infection, which is one reason why the virus continues to spread and cause mild-to-moderate acute infection and more serious long term consequences. It would be best to correct this language in the introduction to avoid confusing readers.

Response: We agree and thank the Reviewer for this critical comment and have adjusted the language in the introduction to avoid confusing readers about the nature and impact of mRNA vaccines, and how our therapeutic differs from its mechanism of action and timing (with respect to abrogation/ treatment of infection). This portion of the Introduction now reads:

“In recent decades, many novel viruses originating in the animal kingdom have been spreading in the rapidly growing human population. Among them, the highly contagious severe acute respiratory syndrome coronavirus 2 (SARS-CoV-2) which caused the COVID-19 pandemic has claimed millions of lives worldwide and overwhelmed global healthcare systems for several years¹. During the infection process, the viral ‘spike’ protein binds the angiotensin-converting enzyme 2 (ACE2) receptor that is expressed ubiquitously in human cells²⁻⁴. The mRNA vaccines developed against SARS-CoV-2 have proven to be an effective strategy against severe disease and death in infected patients, and whilst more effective against early viral strains; their protection

against new viral variants becomes less efficient, unless they are based on new viral strains⁵.

Recombinant hACE2 and virus-specific antibodies from convalescent patient plasma have been explored as decoys for SARS-CoV-2 inhibition, but the risk of blood-borne disease, immune rejection and demand of logistical expertise to purify and manufacture biological (blood) products, limits broad availability and extends higher costs^{6,7}. In general, the specificity of antibodies towards target proteins are conferred by the conformations that the complex antibody active sites adopt and their surface interactions. Despite their large sizes, numerous antibodies (convalescent plasma to in situ vaccine-derived antibodies) have gradually diminishing activities against viral variants/mutations^{8,9}."

7. It was challenging to discern what was a new result versus a reference from the literature in the first paragraph of the Results & Discussion section (discussing Figure 1). It would be useful to further clarify what part of the stated binding results and viral inhibition results are new in this work versus coming from the literature.

Response: We appreciated this comment and acknowledge the identified lack of clarity. We have edited the "Results and Discussion" first para for clarity of new/ previous results. (please see Reviewer 1 Comment 1.) That section now reads, with more clarity:

"The published atomistic binding structure of Spike-RBD coupled to ACE2⁴⁴ provided a means to understand the interaction between the ACE2 α 1 helix domain and Spike-RBD, which was mutated to enhance its binding to RBD (Fig. 1). These mutated peptides demonstrated that a more stable α -helix that aided in binding to RBD, with nM inhibition of SARS-CoV-2²³, akin to other ACE2 domain mimicry strategies recently investigated⁴⁵⁻⁴⁷.

Enhanced binding with self-assembling sequences

Initial attempts in peptide design involved isolation of a short Spike Binding Peptide (SBP) sequence (IEEQAKTFLDKFNHEAEDLFYQS) from ACE2 α 1¹⁷. This short sequence, termed SBP1, was the first bioactive domain that was tested. The SAP domains were conjugated to SBP1 with 2 alternatively charged SAP terminal residues, glutamic acid (E)^{24,48} and lysine (K)^{30,31,49} (Fig. 2A). We observed a significant (~ 70 %) inhibition of pseudovirus by SBP1 from media control (Fig. 2B), as reported previously^{17,18}. E-flanked SAP termed E1 (E-SLSLSLSLSL-E) was conjugated to SBP1 (ESBP1) which inhibited pseudovirus to a similar degree, while K-flanked SAP termed K1 (K-SLSLSLSLSL-K) conjugated to SBP1 (KSBP1) had a significant decrease in inhibitory effect (Fig. 2B). A secondary B.1 lineage live virus plaque formation assay was performed with SBP1 and ESBP1 to confirm if the self-assembling domain could assist in viral inhibition. In the live virus assay, counts of infectious plaques confirmed a significant improvement in viral inhibition by ESBP1 relative to SBP1 alone (Fig. 2C)."

8. In the illustrations of Receptor Binding Domains (figures 1, 5), it would be useful to use an arrow or other indicator to point out the critical binding pocket for non-experts.

Response: We would like to thank the Reviewer for this comment. We have added re-rendered illustrations to clearly point out the critical binding pocket (using a distinct color scheme that is reflected in all the figures -specifically in Figures 1 and 5, Supplementary Figure 1). Further, we have added language outlining the residues identified as active residues for the ligand and binding

pocket of the receptor in the captions for Figure 1 & 5. Figure 1 was re-rendered (for improved resolution) to highlight these residues as well more effectively.

9. It is unclear how the stated IC₅₀ value of ~1 μ M was arrived at from Figure 3G, as the response did not reach saturation in the data shown. Looking at the plot, the concentration at which half of the plaque formation was inhibited appears somewhere between 1-10 μ M. The calculation of the IC₅₀ should be clarified.

Response: We appreciate the Reviewer's critical comment on this IC₅₀ value. To this end, we have revised that portion of the Results and Discussion to be more exact. We have clarified this IC₅₀ and more specifically quoted the value, vs the approximate earlier mentioned. Please note that the calculations of the IC₅₀ are present in the methods. That section now reads:

"). Like ESBP2, ESBP3 showed excellent cytocompatibility with human alveolar epithelial A549 cells and dose-dependent antiviral inhibition with a calculated IC₅₀ of 2.5 μ M (SI Fig. 1, Fig. 3G). ESBP3 showed rapid clearance from the circulation (< 4 hours) after IV bolus administration – characteristic of IV (peptide) therapeutics (SI Fig. 2)."

10. Some of the results for the viral plaque assays are reported as dose-response data, while others are a comparison of different peptides, presumably at a fixed dose. The concentration of each peptide should be indicated in the figure or caption, e.g in Figure 2C.

Response: We appreciate the Reviewer's comment on how we can improve the clarity of the figure presentation. We have carefully clarified the concentration in the figure captions to ensure there is no ambiguity (especially in Figure Figures 2 and 4).

11. The x-axis labels in Fig 4J are misaligned with the bars, making it difficult to know which bar is which.

Response: We appreciate these comments. We have carefully edited this and ensured the PDF conversion software does not reformat the image incorrectly as prior. This is reflected in the revised Figure 4.

12. Several of the graphs in the SI figures have extremely tiny fonts on the axes and are difficult to read. Others are low resolution (e.g. SI Fig 2D, E). These fonts should be made legible.

Response: We appreciate the Reviewer for this comment to enhance the clarity of the manuscript and Figures' presentation. We have made all (and SI) Figures more legible and reformatted them at higher resolution in Main Manuscript and SI Figures.

Reviewer #2 (Remarks to the Author)

1. Spike-binding peptides and anti-SARC-CoV2 SAPS: Why was the SAP conjugated to the N terminus of the spike binding peptides? What happens when the SAP are conjugated to the C terminus?

Response: This is an interesting question posed by the Reviewer that we have not been investigated. We have conventionally performed N-terminal modifications for their: i) known uniquely reactive sites, ii) conjugation stability and iii) efficacy from prior data from our group and others. We did not characterize constructs for C vs N termini modification, and conventionally pursued the common N-terminal modification as premised by previous works. We believe this may not be a major focus of the current work, as a detailed study of other termini/ side chain modifications is out of scope for the current work where we pursued canonical modification schemes.

2. TEM: Some structural characterization beyond the micrograph in the TEM in Figure 3/4 should be requested, such as 2D avgs for ESBP3, E1, ESBP3+E1, and ESBP3+RBD. There is currently no higher resolution structural validation that ESBP variants+E1 mixtures actually form fibrils, despite the representative CryoEM movie in 4I. Further, how were the 1:3 and 1:7 ratios determined?

Response: We thank the Reviewer for this important comment. We have been attempting 2D and 3D reconstructions of ESBP3, E1 combinations and ESBP3+RBD for 3 years now in collaboration with Mario Borgnia (NIEHS) and BVV Prasad (Baylor College of Medicine) with no success. Many other groups too have attempted this with lack of success. This is because the peptides, while forming fibers visualized in AFM, do not have fixed periodicity. Thus, class averaging has been a significant challenge even with these experts, notwithstanding several months (years) of imaging and attempts. Notwithstanding, we have presented in SI Figure 3 the work to date with illustrates this exact challenge – with some preliminary success specific to E1. To circumvent this, ongoing work (as presented in Supplementary Figure 4 is exploring the use of solid-state NMR to probe “nearest-neighbor” interactions to refine computational models (in collaboration with co-author Anant Paravastu (GaTech)). While this may afford some higher resolution for local interaction, higher resolution EM reconstruction are currently impossible due to the challenges with the irregular twists (periodicity) of the self-assemblies. We believe next avenue in determining 2D/3D class averaged structures to be outside the scope of the current presented study – both due to the inability of these reconstructions (with the current technologies available) and scope of the work. Notwithstanding, we are pursuing these in future studies using newer, and upcoming Cryo-EM imaging capabilities and image processing techniques. With respect to the ratios: these were chosen based on preliminary computational modeling presented in the manuscript and which we have further clarified in the Results and Discussion which now reads:

“Examination of the self-assembled β -sheets

MD simulations:

To better understand the structure of these mimics, various self-assembled peptide structures were analyzed with molecular dynamics (MD) simulations. First, we examined whether parallel or anti-parallel orientations take part between E1 self-assembled in β -sheet planes. The calculated MM-GBSA free energies (see Methods for details) of binding between E1 and its surrounding β -sheet plane showed that E1 preferentially assembles into an anti-parallel β -sheet. The calculated free energies were $\Delta G = -72.58$ kcal/mol, -121.23 kcal/mol, and -124.76 kcal/mol for all parallel, planar antiparallel, and all antiparallel, respectively (Figs. 4A-C). This suggests a preponderance towards fully antiparallel self-assembly, but not exclusively anti-parallel arrangement; statistical distributions of peptide configurations within the self-assembled fibrils are controlled by their free energies. The MD simulations revealed variable non-periodic twists within the fibrils, analogous to variable twist rates observed within an E1 fiber during Cryo-EM image collection. This

variability in twist limited our ability to resolve the cryo-EM model at adequate resolution, but it did allow us to observe spacing between the monomers and an approximate multimeric structure similar to our models (SI Fig. 3).

Motivated by the improved structure of ESBP3 by site specific mutations, we investigated, by MD simulations, the predicted reduction in steric hindrance by incorporating just the self-assembling domain E1 into ESBP3 fibers. We generated models of 16-mer fibers containing a 7:1 molar ratio and a 3:1 molar ratio of E1 and ESBP3, and pure ESBP3 (Figs. 4D-F). The E1 fibers containing no bioactive domain were hypothesized to serve as a spacer molecule for the multidomain ESBP3 fibers. Solvent Accessible Surface Area (SASA) measurements showed an inverse relationship between the ESBP3 molar proportion and the area of mimic accessible. This agrees with the calculated average SASA of 6,635 Å²/mimic for 7:1 E1:ESBP3, 4,095 Å²/mimic for 3:1 E1:ESBP3, and 2,063 Å²/mimic for all ESBP3 (Figs. 4D-F). This warranted further testing through solid state NMR interactions and ultimately live virus inhibition.

NMR experiments:

When the E1, ESBP3, and E1:ESBP3 in a 3:1 molar ratio assembly were examined with NMR, we observed centrifuge pellets and concomitant solid-state NMR signals from the samples of E1 and 3:1 E1:ESBP3. This indicates their assembly form organized secondary structures typical for a peptide nanofiber assembly, vs ESBP3 (no pellet / no solid-state NMR signal). Comparing the spectra collected for the E1 and 3:1 E1:ESBP3 assemblies, both exhibited peaks with linewidths of ~ 2 ppm, some overlapping peaks, and some peaks that are unique to the 3:1 E1:ESBP3 sample (Fig. 4G). The linewidths were typical for ¹³C natural abundance spectra collected for amyloid fibrils of designer peptides⁵³. While natural abundance ¹³C NMR is typically insufficient to perform spectral assignments (correspondence between NMR peaks and ¹³C sites), partial ¹³C peak assignments were possible because known ¹³C chemical shift ranges for E, S, and L residues within the E-(SL)₆-E self-assembling domain in both peptides⁵⁴. Many peaks observed in the E1 spectrum align with the peaks from 3:1 E1:ESBP3, as could be anticipated since both peptides assemble into β-sheets and the E1 peptide is the most abundant peptide in both samples (Fig. 4G). The strongest signal was at 28 ppm, indicating that the presence of the ESBP3 peptide affects the assembled nanofiber structure, consistent with the SBP3 peptide domain associated with 25% of molecules in the 3:1 E1:ESBP3 sample (Fig. 4G). The signals indicate that the SBP3 domain was incorporated into the assembly, but also suggest that a portion of the residues in the SBP3 domain may be integrated into the β-sheet assembly.

Given the lacking solid-state NMR signal indicating the assembly of ESBP3 peptide relative to the E1:ESBP3 mixture, we sought to probe soluble monomeric and oligomeric peptide assemblies using solution NMR. To this end, 1D ¹H solution NMR spectra of 1 mg/mL solutions of E1, ESBP3, and 3:1 E1:ESBP3 mixtures were acquired and compared (Fig. 4H). ESBP3 peptide exhibited a strong solution NMR signal (Fig. 4H), in stark contrast to its undetectable solid-state NMR signal (Fig. 4G). E1 peptide and 3:1 E1:ESBP3 mixtures exhibit very weak solution NMR signal (Fig. 4H) and corroborating solid-state NMR signals suggesting these peptides form nanofiber structures (Fig. 4G). Notably, the E1 peptide and 3:1 E1:ESBP3 mixtures do exhibit some signal in solution NMR experiments, albeit a different spectral profile from isolated ESBP3 peptide. Several NMR peaks in solution NMR spectra of ESBP3 exhibit narrow linewidths (< 0.1 ppm full width half maximum) consistent with unassembled, free peptide monomers in aqueous

solution. Broad NMR peaks with wider linewidths (~ 0.2 ppm) were also present in the ESBP3 solution NMR spectra, indicating the presence of soluble, oligomeric, peptide assemblies (Fig. 4H) – these fibrillar structures were further visualized in cryo-EM (Fig. 4I). To determine whether the broad NMR peaks correspond to soluble peptide assemblies, we performed 1D ¹H Carr-Purcell-Meiboom-Gill (CPMG) solution NMR experiments⁵⁵, which utilize optimized pulses to filter out NMR signals corresponding to larger molecules with unique relaxation properties (specifically, small T₂ relaxation values for oligomers with large molecular weights)⁵⁶. The ¹H CPMG experiments indeed illustrated the presence of both soluble, oligomeric assemblies and monomeric peptides of ESBP3, as shown by the loss of NMR signal corresponding to broad NMR peaks and the relatively unaffected NMR signal of the narrow NMR peaks as a function of the CPMG relaxation filter (SI Fig. 4).

Taken together, solid- and solution-state NMR indicate that the SBP3 domain can be incorporated into a self-assembled nanostructure formed by the E1 domain. For the ESBP3 peptide solution, which contains E1 and SBP3 in every peptide molecule, the SBP3 interferes with E1 assembly, resulting in soluble aggregates that can be detected by solution NMR, but did not pellet via ultracentrifugation. The solid-state NMR data showed that ESBP3 was incorporated into a nanostructure that could be pelleted when ESBP3 was co-assembled with E1 in the 3:1 E1:ESBP3 sample (Figs. 4G-H). Comparison of the spectrum from 3:1 E1:ESBP3 to the spectrum of an E1 assembly indicates that SBP3 affects the assembled structure, but further research is necessary to pin down the structural details. Thus, despite a degree of efficacy we suggest that there may be room for improvement in the self-assembling peptide design. ESBP3 is limited in its ability to form β-sheets and fibrils alone (Figs. 3E, 4I), but it is stabilized into a fiber by doping with E1.”

3. Virus inhibition: How were the relative concentrations between peptide and fibril normalized when measuring viral inhibition (4J)? It's unclear how the self-assembly of the fibril actually enhances viral inhibition relative to free non-assembled peptide, as the axes formatting is weird.

Response: We thank the Reviewer for mentioning that the previous presentation lacks image and concentration clarity, as addressed in Reviewer 1 comment 10. We have revised the Manuscript and Figures (and captions) to improve clarity, and we detail the tested concentrations. Further, the choice of viral titer load was based on previous reports and standard assays perform to-date, as cited and detailed in the methods. For the peptides: based on CD, IR and EM imaging the peptides assemble into fibers and entangle into a gel at mM concentrations. In ssNMR experiments we see much less free peptides. Thus, we report the total concentration of peptide, noting that majority of the peptide is in the condensed phase, as do our ssNMR, CD and IR measurements of the supernatant suggest.

4. Antiviral activity of fibrils: At the moment the manuscript first focuses on comparing antiviral activity of each peptide variant without followed by modeling of fibril assembly. The authors should further emphasize the utility of antiviral fibrils vs peptides (e.g. comparison of half-life, shelf-life, serum stability, clearance, binding affinity, virus neutralization).

Response: We appreciate the Reviewer's comment on the focus of the work. We enhanced the “Results and Discussion” to include these critical features complemented by improved figures for the half-life, stability (Supplementary Figure 1-2) and binding affinity (Figure 3-4). Complimenting

the discussion / response to Reviewer 1, please see Critique 4/5 as well. Further addressing this, we have improved that portion of the Results and Discussion and included new SI Videos. This section now states:

“Live virus inhibition with self-assembled peptide mimics

Live virus plaque inhibition assays against the B.1 strain was performed on E1, SBP3, ESBP3, and E1:ESBP3 in a 3:1 molar combination to determine their therapeutic efficacy. While no major differences were determined between E1, SBP3, and ESBP3, the 3:1 molar combination of E1 and ESBP3 had significantly higher viral inhibition (Fig. 4J). This enhanced viral inhibition in diluted fibrils could have several possible origins, such as specificity to Live virus/ Spike by targeting RBD (of ESBP3/ E1:ESBP3) or non-specific ionic interactions of poly-anionic E1. However, spacing of ESBP3 domains with just E1 SAP domains, as suggested by NMR, promoted better presentation of the binding domain to RBD.

To understand these experimental inhibition observations, we performed MD simulations of individual peptides and their assemblies coupled to RBDs in the Alpha (B.1.1.7) and Omicron (B.1.1.529) variants. Starting with the individual peptides binding to the Alpha receptor, the E1 domain moved away from the canonical binding pocket (Fig. 5A). SBP3 migrated around the binding pocket, while rotating and translating away from the traditional residues associated with binding to ACE2 (Fig. 5B), but ESBP3 showed more efficient with a limited migration from the canonical binding pocket (Fig. 5C). ESBP3 also demonstrated higher coupling energy with an average of -68.4 kcal/mol, vs. -36.0 kcal/mol and -38.8 kcal/mol for SBP3 and E1 respectively. For the Omicron RBD, E1 localized around the positively charged RBD (Fig. 5D, SI Fig. 5) with a similar coupling energy of -32.1 kcal/mol. SBP3 showed a decreased coupling energy against Omicron vs. Alpha (Fig. 5E) at -34.0 kcal/mol, but ESBP3 again showed improved binding relative to E1 and SBP3 at -96.4 kcal/mol (Fig. 5F). MD simulations against a widespread variant (B.1.529) suggested ESBP3 and the E1 would have improved binding against the receptor binding motif (RBM), likely due to the increased positive electrostatic surface potential localized on the Omicron RBD (Figs. 5E-G, SI Vid. 1-3)⁵⁷. This data supports the hypothesis that combining the SBP3 Spike Binding Peptide with the E1 SAP can enhance its binding to the receptor (Fig. 2, Fig. 5A-F, SI Fig. 5).

We simulated fibrils with the 7:1 E1:ESBP3 molar ratio 16-mer, the 3:1 E1:ESBP3 molar ratio, and all ESBP3 combinations coupled to the Alpha and Omicron variants. For Alpha variant, we saw average coupling energies increased with increasing relative ESBP3 concentration, with values of -35.6 kcal/mol, -50.4 kcal/mol, and -60.9 kcal/mol for 7:1 E1:ESBP3, 3:1 E1:ESBP3, and all ESBP3 respectively. For the Omicron variant, there was a less pronounced difference in energy, with values of -57.0 kcal/mol, -54.2 kcal/mol, and -69.8 kcal/mol for 7:1 E1:ESBP3, 3:1 E1:ESBP3, and all ESBP3 respectively (Figs. 5H, I, K, L, SI Fig. 6, SI Videos 1-3). While the 7:1 E1:ESBP3 showed selective binding to the canonical binding pocket (Figs. 5G, J, SI Video 3), as the molar ratio of ESBP3 increased, the selectivity of binding started to decrease at full occupancy of the bioactive domains (Figs. 5H, K, SI Videos 1-3).

Coupling energy of the whole fibril alone did not explain our live virus observations that spacing the multidomain peptides would yield a more effective antiviral (Figs. 5G-L, SI Fig. 6). At full occupancy of the fibril by the bioactive domains, various other coupling regions bound to RBDs (Figs. 5I, L, SI Video 3). Overall, our MD simulations of fibrils with different ratios of E1:ESBP3

showed that as the proportion of ESBP3 increases, solvent accessible surface area (SASA) decreases, leading to disordered presentation of the bioactive domains⁵⁸ (Figs. 4E-F, 5G-L). This, along with our live virus data (Fig. 4J), suggests a hypothesized optimization⁵⁹⁻⁶¹ for future hybrid constructs by appropriate combinations of pure self-assembling peptide and multidomain peptide monomers is warranted as evidenced by the significantly enhanced binding and live virus inhibition by E1:ESBP3.”

5. Binding affinity predictions: The affinity in 3C was all predicted. Is there any validation experimentally that the actual binding affinity of the SBP variants reflect the in silico modeling? Further, could the authors test whether the predicted binding orientations of SBP3 and ESBP3 peptides bind in the predicted orientations and whether the fibrils form in the predicted orientations (Fig 5)?

Response: This is an important comment of the Reviewer. To this end, we have in silico predicted binding for each construct that was tested, and their presentation (especially in Figure 4J) was clarified. With respect to imaging of peptides / fibers binding to RBD or Spike - as we have noted in the manuscript (and please see detailed response to Reviewer 2 Critique 2)– due to the lack of periodicity, it is impossible to develop a high(er) resolution reconstruction of the viral-peptide (nanofiber) interaction. Notwithstanding, we have presented the ssNMR data and EM data that support the supramolecular self-assembly and binding as presented (see Reviewer 2, Critique 2)

6. The authors claim larger self-assemblies can wrap the virion surface, although there is no structural data showing that the fibrils are wrapping themselves around the virion or neutralizing in some other way and are missing a citation with this claim in their introduction.

Response: This is an important and critical comment from. We acknowledge that, due to the heterogeneity of lengths, bending and aperiodic nature of the fibers, we cannot elucidate a full structure of this interaction. However, as shown in Figure 3 E and F and Figure 4 I– the fibers are in fact flexible, and can potentially interact with RBD/Spike on virions. However, we appreciate this comment and have revised the manuscript to limit this claim of “binding and wrapping the virion” given the challenges with imaging (detailed above). We have edited the manuscript to better substantiate the results we currently have presented. We have additionally enhanced the introduction with citations for this as recommended (see reviewer 1 comment 4 & 5).

7. Self assembly process: Some details explaining the self-assembly process of E1 with ESBP3 could be helpful such as buffer conditions, assembly time and temperature, assembly order with virion / pseudovirus, etc.

Response: We thank the Reviewer for the comment. We have added these details to the “Methods” section.

This section now reads;

“Peptides were prepared on a LibertyBlue solid phase peptide synthesizer (CEM, Matthews, NC) using standard Fmoc chemistry, (1:4:4:6 resin:amino acid:1-[bis(dimethylamino)methylene]-1H-1,2,3-triazolo[4,5- b] pyridinium 3-oxid hexafluorophosphate:diisopropylamine, 0.1 mM scale) and all peptides were N-terminally acetylated (achieved with a 1:3 ratio of acetic anhydride to diisopropylethylamine in dichloromethane) and C-terminally amidated (low loading Rink amide resin, 0.36 mmol/gram). After synthesis, the crude peptides were cleaved with 0.25 mL each of

MilliQ water, triisopropylsilane, and 3,6-dioxa-1,8- octanedithiol (DoDT) and 9.25 mL trifluoroacetic acid for 30 min at 37 °C^{32,33}. Post cleavage the crude peptides were filtered with a fritted column and triturated with cold ether. The ether and peptide mixtures were vortexed and centrifuged, the ether was then decanted, and the crude peptides were left to dry overnight. The following day the crude peptides were dissolved at approximately 1 mg/mL in Milli-Q water (pH ~ 7.0). These peptide solutions were dialyzed (Spectra/Por S/P 7 RC dialysis tubing, 2 kDa MWCO) against deionized water for 3 days (water changed 3x daily, 1:1000 peptide to reservoir volume). The peptides were then frozen and lyophilized to obtain the final peptides. The expected molecular weights of the peptides were confirmed with LC/MS. Peptides were reconstituted at room temperature at 10 mg/mL (or stated molarities) in saline with dissolution in <1minute, and stored at 4C till used^{49,74,84}. Mixtures of peptides were formulated at equimolar (or stated) concentrations by mixing solubilized formulations, brief vortexing, and sonication prior to use.”

8. Small improvements:

8.1 Figure 2B: The controls for E1 and K1 inhibition of SARS pseudovirus are missing.

8.2 Figure 4J: The x axis layout is confusing

8.3 Figure S1F: There is actually no comparison between ESBP2 to SBP2, as only data for ESBP2 is shown.

8.4 SUPPLIER is missing for the 400-mesh EM grids.

Response:

8.1: We did not test /screen any peptides that did not have a spike binding domain in initial studies, however, we did test the self-assembled version of K1 and E1 vs pseudovirus. Noting the critical differences in pseudo- and live virus structure, but preserved specificity of RBD (recapitulated in pseudovirus assays) – this assay was used as a screening tool to determine which SAP backbone would be best suitable to carry forward (focus of Figure 1), while subsequent studies (Figure 2-4), which involved livevirus did test unfunctionalized SAP alone Further, during the study, SBP1 alone was shown to not be efficacious against live-virus – as detailed in the manuscript – and thus did not warrant re-investigation of E1 till it was used as a control in Figure 5.

8.2: Please see reviewer 1 comment 11 – we have corrected this to improve clarity

8.3: See comment for 8.1 above – we screened peptides with Spike binding domains, and only when optimized screen (ESBP3, Figure 4-5) screened E1 and combinations.

8.4: We have updated the supplier for the product.

Reviewer #3 (Remarks to the Author):

(1) Regarding the design scheme, the team started with testing existing designs SBP1, SBP2 as fusions with SAP: ESBP1 and ESPB2. HADDOCK-derived complexes of SBP2 with RBD were subject to computational Ala-scan mutagenesis to identify hot-spot residues. Other residues were truncated to allow for more room in the self-assembled construct. This approach seems reasonable but for scientific completeness, the approach needs to be described in more detail - perhaps in the supplement. Details on the HADDOCK protocol, number of poses evaluated by ala-scanning, cutoff for hotspot/non-hotspot residues are not presented. Finally, the protocol for choosing replacement residues was not described. Whether it was rational or computational - the motivation behind the SPB3 changes should be described.

Response: We would like to thank the Reviewer for this valuable comment. We have enhanced the methods include details on the HADDOCK protocol, number of poses evaluated by alanine-scanning, cutoff for hotspot/non-hotspot residues. We have added our rationale and method for choosing replacement residues was not described leading to SBP3 design. These updates are presented in the “Evaluation of SBP1 and SBP2” and “Development of SBP3” sections in the methods. That section now reads:

“Evaluation of SBP1 and SBP2

SBP1 sequence (IEEQAKTFLDKFNHEAEDLFYQS) was derived from the interacting regions of ACE2 by Pomplun et al.¹⁷. SBP2 sequence (SALEEQLKTFLDKFMHELEDLLYQLAL) was derived by Karoyan et al.²³. The SWISS-MODEL Expasy web server⁶⁵ was used to create structural models of peptides presented in prior literature and which were mutated from ACE2 α 1 helix residues 19-45 using homology-based modeling of the SARS-CoV-2 ACE2-RBD complex PDB ID: 6M0J)⁶⁶. HADDOCK molecular docking runs between each peptide and spike RBD were executed using default docking parameters, and six complexes from the cluster were selected based on predicted energy of interaction for further analysis and comparison to the ACE2-RBD complex^{51,67}. A predicted binding energy and dissociation constant (K_D) were generated under standard conditions for each complex using the PRODIGY web server⁶⁸, and Gibbs Free Energy values were predicted using the BAaS BUDE Alanine Scanning web server⁵². The latter web server also provided insights into the effect of a synonymous mutation at each amino acid (R \rightarrow Ala) in the peptides on binding, enabling the identification of crucial residues which made a significant contribution based on their differential Gibbs Free Energy ($\Delta\Delta G$) values. Residues with a $\Delta\Delta G$ higher than 0.718 kcal/mol were considered most impactful, and residues with a $\Delta\Delta G$ higher than 0.239 kcal/mol were considered essential⁶⁹.

Development of SBP3

The overall design strategy was guided by the SARS-CoV-2 Spike-RBD bound ACE2 complex (PDB ID: 6M17 and 6M0J)^{70,71}. SBP3 (QYKTYIDKNNHYAEDERYK) was developed by comparing the BAaS BUDE outputs for SBP1 and SBP2 peptides. The residues that had large, positive $\Delta\Delta G$ values were scrutinized and considered for the development of the novel constructs. A new peptide sequence was created from a combination of these significant residues, which included residues conserved from the original ACE2 α 1 helix and residues from the tested peptides. Two amino acids had comparable $\Delta\Delta G$ values at three positions in the peptide (10, 15, and 17), necessitating the creation of $2^3 = 8$ variant sequences.

Each sequence was validated using the protocol outlined in the “Evaluation of SBP1 and SBP2” section above. Secondary analysis was performed using the BeAtMuSiC web server⁷². In addition to assessing the effects of an R \rightarrow Ala mutation, the BeAtMuSiC webserver assesses the effects of every possible synonymous mutation at each amino acid residue. The $\Delta\Delta G$ values obtained from this analysis were used to create a variant peptide, but this peptide performed poorly in the evaluation and thus omitted from this work⁷³.

Development of ESBP1, KSBP1, ESBP2, and ESBP3

The self-assembled analogues were developed by conjugating the SBP mimics (SBP1, SBP2, SBP3) into previously published sequences E1 (ESLSLSLSLSLE)^{24,48} or K1 (KSLSLSLSLSLK)^{28,74} using a glycine spacer. The existing self-assembling peptide K-(SL)₆-K or E-(SL)₆-E backbone was conjugated to an SBP mimic with a glycine spacer yielding X-(SL)₆-

X-SBPy - where X is E or K and y is SBP1/2/3, onto which a peptide could be added at the C-terminus. All peptides were dissolved in pharmaceutical grade saline (0.9% NaCl)."

(2) Molecular dynamics is the computational tool used to interpret biophysical observations at a molecular level. Initially, it is used to differentiate between all-parallel, planar antiparallel and all-antiparallel models of the E1 assembly. This is tricky - but reasonable in the context of the current project. Structures were generated in pymol and subjected to fast-relax in rosetta. Implicit solvent, langevin dynamics simulations are performed. Nonbonding interal energetics calculated as well as protein-solvent interactions reported. It is difficult to evaluate the quality of the calculations without more details. Specifically:

2.1 Is What are the criteria for convergence (energy/RMSD/geometry)? 400 ns enough for equilibration and production trajectory analysis? How many replicas were run for each calculation? For a system of this size, one is likely not enough. Justification and detailed presentation of simulation results should be included in the manuscript and supplement.

2.2 Several structures of SAP are available, how do the E1 models compare? In the end, the all-antiparallel solution is the one nature chooses as well, so what is the motivation for the MD calculation?

Response: 2.1 - We appreciate this astute criticism and have revised our Results and Discussion to include: criteria for convergence (energy), a detailing of how 400 ns is enough for equilibration and production trajectory analysis, repeats of 3 replicas (SI Figure 6 and Methods) were run for each calculation as details in the in the Methods and Results and Discussion sections.

2.2 – The Reviewer has carefully pointed out that other SAP models exist. Our simulations suggest that within assembled fibers, there may be more than 1 orientation, but a preponderance towards anti-parallel. We have revised the Results and Discussion to tie together consistent observations between our Cryo-EM studies and these MD simulations. Namely, the twist observed for our fully parallel and planar antiparallel orientations that was not observed with planar antiparallel orientations for our MD simulations, and the variable twist observed in the Cryo-EM examinations of the fibrils that prevented full resolution of the structure. We discuss this in the first paragraph of the “Examination of the self-assembled β -sheets” section of the results.

Revised Methods:

“Molecular dynamics simulations

MD simulations were performed using NAMD⁷⁶ with the CHARMM36 protein force field⁷⁷. The simulations were performed in a physiological solution with 0.15 M NaCl. The cutoffs of van der Waals (vdW) and Coulombic coupling were 10 Å. The particle mesh Ewald (PME)⁷⁸ method was used for the evaluation of long-range Coulombic interactions and to increase the efficiency of the simulations. The simulations were performed in the NPT ensemble (P = 1 bar and T = 310 K). The systems were simulated for 400 ns using the Langevin dynamics with a damping constant of 1 ps⁻¹ and a time step of 2 fs, and repeated 3 times to ensure consistency.”

Revised Results and Discussion:

“Examination of the self-assembled β -sheets

MD simulations:

To better understand the structure of these mimics, various self-assembled peptide structures were

analyzed with molecular dynamics (MD) simulations. First, we examined whether parallel or anti-parallel orientations take part between E1 self-assembled in β -sheet planes. The calculated MM-GBSA free energies (see Methods for details) of binding between E1 and its surrounding β -sheet plane showed that E1 preferentially assembles into an anti-parallel β -sheet. The calculated free energies were $\Delta G = -72.58$ kcal/mol, -121.23 kcal/mol, and -124.76 kcal/mol for all parallel, planar antiparallel, and all antiparallel, respectively (Figs. 4A-C). This suggests a preponderance towards fully antiparallel self-assembly, but not exclusively anti-parallel arrangement; statistical distributions of peptide configurations within the self-assembled fibrils are controlled by their free energies. The MD simulations revealed variable non-periodic twists within the fibrils, analogous to variable twist rates observed within an E1 fiber during Cryo-EM image collection. This variability in twist limited our ability to resolve the cryo-EM model at adequate resolution, but it did allow us to observe spacing between the monomers and an approximate multimeric structure similar to our models (SI Fig. 3.).

Motivated by the improved structure of ESBP3 by site specific mutations, we investigated, by MD simulations, the predicted reduction in steric hindrance by incorporating just the self-assembling domain E1 into ESBP3 fibers. We generated models of 16-mer fibers containing a 7:1 molar ratio and a 3:1 molar ratio of E1 and ESBP3, and pure ESBP3 (Figs. 4D-F). The E1 fibers containing no bioactive domain were hypothesized to serve as a spacer molecule for the multidomain ESBP3 fibers. Solvent Accessible Surface Area (SASA) measurements showed an inverse relationship between the ESBP3 molar proportion and the area of mimic accessible. This agrees with the calculated average SASA of $6,635 \text{ \AA}^2/\text{mimic}$ for 7:1 E1:ESBP3, $4,095 \text{ \AA}^2/\text{mimic}$ for 3:1 E1:ESBP3, and $2,063 \text{ \AA}^2/\text{mimic}$ for all ESBP3 (Figs. 4D-F). This warranted further testing through solid state NMR interactions and ultimately live virus inhibition.”

3. This is overall a successful project of interest to the community if the above concerns are considered. I would just add one more general point - I could not find a hypothesis for why fusing a bioactive peptide and a self-assembling one would result in a better anti-viral. I can imagine two mechanisms - Confinement reduces the available volume, dampening dynamics and pushing the folding funnel for the helix toward a folded, helical state. In this case, similar increase in affinity could be expected for higher ratios of E1 to ESBP3. Alternatively, the increased efficacy is not affinity, but avidity. Either mechanism would have different implications for design and formulation.

Response: We thank the reviewer for this excellent comment and have augmented the “Abstract”, “Introduction” and “Results and Discussion” sections to reflect the potential hypotheses as to why E1 may enhance ESBP3 binding. Our ongoing simulations compliment the Reviewer’s hypothesis “**Confinement reduces the available volume** “ and have augmented specifically the Results and Discussion section with details and potential mechanisms; additionally see responses to Reviewer 1 comment 1 and Reviewer 2 comment 5. The direct hypothesis of bioactive domain presentation (quantified by solvent accessible surface area) in the last paragraph of the “Live virus inhibition with self-assembled peptide mimics” section of the results.

This section now reads:

“Live virus plaque inhibition assays against the B.1 strain was performed on E1, SBP3, ESBP3,

and E1:ESBP3 in a 3:1 molar combination to determine their therapeutic efficacy. While no major differences were determined between E1, SBP3, and ESBP3, the 3:1 molar combination of E1 and ESBP3 had significantly higher viral inhibition (Fig. 4J). This enhanced viral inhibition in diluted fibrils could have several possible origins, such as specificity to Live virus/ Spike by targeting RBD (of ESBP3/ E1:ESBP3) or non-specific ionic interactions of poly-anionic E1. However, spacing of ESBP3 domains with just E1 SAP domains, as suggested by NMR, promoted better presentation of the binding domain to RBD.

To understand these experimental inhibition observations, we performed MD simulations of individual peptides and their assemblies coupled to RBDs in the Alpha (B.1.1.7) and Omicron (B.1.1.529) variants. Starting with the individual peptides binding to the Alpha receptor, the E1 domain moved away from the canonical binding pocket (Fig. 5A). SBP3 migrated around the binding pocket, while rotating and translating away from the traditional residues associated with binding to ACE2 (Fig. 5B), but ESBP3 showed more efficient with a limited migration from the canonical binding pocket (Fig. 5C). ESBP3 also demonstrated higher coupling energy with an average of -68.4 kcal/mol, vs. -36.0 kcal/mol and -38.8 kcal/mol for SBP3 and E1 respectively. For the Omicron RBD, E1 localized around the positively charged RBD (Fig. 5D, SI Fig. 5) with a similar coupling energy of -32.1 kcal/mol. SBP3 showed a decreased coupling energy against Omicron vs. Alpha (Fig. 5E) at -34.0 kcal/mol, but ESBP3 again showed improved binding relative to E1 and SBP3 at -96.4 kcal/mol (Fig. 5F). MD simulations against a widespread variant (B.1.529) suggested ESBP3 and the E1 would have improved binding against the receptor binding motif (RBM), likely due to the increased positive electrostatic surface potential localized on the Omicron RBD (Figs. 5E-G, SI Vid. 1-3)⁵⁷. This data supports the hypothesis that combining the SBP3 Spike Binding Peptide with the E1 SAP can enhance its binding to the receptor (Fig. 2, Fig. 5A-F, SI Fig. 5).

We simulated fibrils with the 7:1 E1:ESBP3 molar ratio 16-mer, the 3:1 E1:ESBP3 molar ratio, and all ESBP3 combinations coupled to the Alpha and Omicron variants. For Alpha variant, we saw average coupling energies increased with increasing relative ESBP3 concentration, with values of -35.6 kcal/mol, -50.4 kcal/mol, and -60.9 kcal/mol for 7:1 E1:ESBP3, 3:1 E1:ESBP3, and all ESBP3 respectively. For the Omicron variant, there was a less pronounced difference in energy, with values of -57.0 kcal/mol, -54.2 kcal/mol, and -69.8 kcal/mol for 7:1 E1:ESBP3, 3:1 E1:ESBP3, and all ESBP3 respectively (Figs. 5H, I, K, L, SI Fig. 6, SI Videos 1-3). While the 7:1 E1:ESBP3 showed selective binding to the canonical binding pocket (Figs. 5G, J, SI Video 3), as the molar ratio of ESBP3 increased, the selectivity of binding started to decrease at full occupancy of the bioactive domains (Figs. 5H, K, SI Videos 1-3).”

Reviewers' Comments:

Reviewer #1:

Remarks to the Author:

All of my concerns from the original review have been addressed in an acceptable way. It does seem that the E1 self-assembling peptide has distinct anti-viral activity in the plaque-forming assays from the specific Spike-binding peptides, but this was mentioned at various places in the revised manuscript so it is acceptable. The new color coding of the simulations in grey/red is especially clear. Since this manuscript describes the development of a viral-neutralizing peptide system with potential for further testing as a preventative for SARS-CoV-2 infection, and all concerns have been addressed, I recommend it for publication.

Reviewer #2:

Remarks to the Author:

The manuscript by Dodd-o and Roy et al. describes the development of a novel self-assembly fibril system to bind and assemble on viral particles. In this revised version of the manuscript the authors have done a very commendable job to address the reviewers' various concerns and raised points have been settled.

Reviewer #3:

Remarks to the Author:

The revision includes expanded methods and data analysis for the molecular dynamics simulations, which as presented, meet the standards of practice in the field. Also, the simulations are now well integrated into the scientific narrative. I have no further issues with this.

A minor but important revision is to increase the size of images in the supplement. Many of the panels are unreadable even when zoomed in to the maximum my browser allows - axes labels are just a few pixels.

Dear Reviewers,

We thank the Reviewers for their encouraging reviews (no edits requested). We have made all the Editorial recommended changes as well.

Sincerely,
Vivek Kumar

REVIEWERS' COMMENTS

Reviewer #1 (Remarks to the Author):

All of my concerns from the original review have been addressed in an acceptable way. It does seem that the E1 self-assembling peptide has distinct anti-viral activity in the plaque-forming assays from the specific Spike-binding peptides, but this was mentioned at various places in the revised manuscript so it is acceptable. The new color coding of the simulations in grey/red is especially clear. Since this manuscript describes the development of a viral-neutralizing peptide system with potential for further testing as a preventative for SARS-CoV-2 infection, and all concerns have been addressed, I recommend it for publication.

Response: Thank you.

Reviewer #2 (Remarks to the Author):

The manuscript by Dodd-o and Roy et al. describes the development of a novel self-assembly fibril system to bind and assemble on viral particles. In this revised version of the manuscript the authors have done a very commendable job to address the reviewers' various concerns and raised points have been settled.

Response: Thank you.

Reviewer #3 (Remarks to the Author):

The revision includes expanded methods and data analysis for the molecular dynamics simulations, which as presented, meet the standards of practice in the field. Also, the simulations are now well integrated into the scientific narrative. I have no further issues with this.

A minor but important revision is to increase the size of images in the supplement. Many of the panels are unreadable even when zoomed in to the maximum my browser allows - axes labels are just a few pixels.

Response: We have increased the size/ resolution of images.